# Hydrodynamic accumulation of small molecules and ions into cell-sized liposomes against a concentration gradient

Hironori Sugiyama [1], Toshihisa Osaki [2,3], Shoji Takeuchi[2,4✉] & Taro Toyota [1,5✉]

In investigations of the emergence of protocells at the origin of life, repeatable and continuous supply of molecules and ions into the closed lipid bilayer membrane (liposome) is one of the fundamental challenges. Demonstrating an abiotic process to accumulate substances into preformed liposomes against the concentration gradient can provide a clue. Here we show that, without proteins, cell-sized liposomes under hydrodynamic environment repeatedly permeate small molecules and ions, including an analogue of adenosine triphosphate, even against the concentration gradient. The mechanism underlying this accumulation of the molecules and ions is shown to involve their unique partitioning at the liposomal membrane under forced external flow in a constrained space. This abiotic mechanism to accumulate substances inside of the liposomal compartment without light could provide an energetically up-hill process for protocells as a critical step toward the contemporary cells.

[1] Department of Basic Science, Graduate School of Arts and Sciences, The University of Tokyo, 3-8-1 Komaba, Meguro, Tokyo 153-8902, Japan. [2] Institute of Industrial Science, The University of Tokyo, 4-6-1 Komaba, Meguro, Tokyo 153-8505, Japan. [3] Kanagawa Institute of Industrial Science and Technology, 3-2-1 Sakado, Takatsu, Kawasaki, Kanagawa 213-0012, Japan. [4] Department of Mechano-Informatics, Graduate School of Information Science and Technology, The University of Tokyo, 7-3-1 Hongo, Bunkyo, Tokyo 113-8656, Japan. [5] Universal Biology Institute, The University of Tokyo, 3-8-1 Komaba, Meguro, Tokyo 153-8902, Japan. ✉email: takeuchi@hybrid.t.u-tokyo.ac.jp; cttoyota@mail.ecc.u-tokyo.ac.jp

The emergence of the early cell-like system compartmentalized by lipid bilayer membrane in the early Earth have drawn much attention[1–3]. Compartmentalization is thought to be indispensable in allowing protocells to retain molecules and to evolve the metabolic system as implemented in contemporary living cells[4]. In addition, compartmentalization is key to resisting parasitic chemical replicators, representing another central issue regarding the steady replication in the prebiotic era[5,6].

As an explanation for the origin of the first generation of such compartments encapsulating functional molecules, a 'super concentration' effect has been reported as a promising clue, by which unexpectedly large number of macromolecules is spontaneously encapsulated during the liposome formation[4,7]. Such liposomes containing macromolecules inside retain their contents even upon strong membrane perturbations[8]. However, once the compartment is formed, the compartment itself could hinder the supply of macromolecules to the inside, and as a result, maintenance, propagation, and development of metabolic system would be limited. Especially, the backwash on the supply of substances is critical to the phospholipid membrane, which is implemented in the current living cells, owing to their low permeability[2,9]. Simpler amphiphiles were expected as the plausible candidates of the earliest membrane molecules, such as alkyl phosphate, alkyl sulfates, fatty acids, and so on, the membrane of which have larger permeability[10]. However, the low permeability of phospholipid membrane must be overcome at the stage of early cell-like system. Contemporary living cells implement cooperative reaction network composed of various membrane proteins involving the consumption of chemical energy sources such as adenosine triphosphate (ATP)[11,12] to overcome this low permeability of the phospholipid membrane. Recently, there are remarkable progress on the prebiotically plausible synthetic pathways of phospholipid[13–17], but the physicochemical insights toward a generation of functional compartment composed of phospholipids is limited.

The problem of the supply to the pre-formed compartment is also a central issue in synthetic biology aiming at the reconstruction of cell-like chemical systems. Since liposomes are widely utilized owing to their similarity to the composition of the compartment of the current cells, the low permeability of the phospholipid membrane is a matter to be handled. Thus far understanding and application of pore-forming mechanics at the membrane have been developed[18–21]. However, the strategy based on the passive diffusion of molecules and ions confronts a limitation of their concentration gradient across the membrane. In other words, the concentration of the substances in the compartment must be smaller than or at most equivalent to the outer solution.

A simple mechanism to accumulate and concentrate molecules and ions from the external environment into the preformed compartment with a phospholipid bilayer membrane would be crucial to develop liposome-type protocells[22,23] in the context of exploration on the origin of life and synthetic biology. Thus far, possible molecular scenarios for the origin of life were elucidated in views of molecular candidates in the early earth[24,25], possible prebiotic synthesis of biomolecules[13–17], and potential environments for the birth of the earliest life[26,27]. Besides, bottom-up construction of model protocells have been partly achieved by the synthetic chemical cell models mimicking important features of contemporary cells[28–32]. However, an abiotic mechanism to accumulate substances into the compartment composed of phospholipid bilayer membrane is still unveiled.

Here we demonstrate that small molecules and ions can be accumulated in preformed cell-sized liposomes against the concentration gradient repeatedly. Notably, the process does not require any membrane proteins but is driven only by physical contact to some surface with an external flow. Besides, although direct observation is the powerful technique to measure the precise dynamics of cell-sized liposomes, for solid discussion on the direct observation of each cell-sized liposomes, statistical data analysis is required[33]. Thus, we here conceive and develop a microfluidics-based automatic observation platform which is termed as Machine-Assisted, Numerous, Simultaneous, and Interactive Observation of Non-equilibrium self-assembly (MANSIONs). MANSIONs is a new tool for conducting precise and reproducible protocol on the investigation of preformed cell-sized liposomes by direct observation.

## Results and discussion

**Accumulation of uranine into a liposome**. Cell-sized liposomes used were composed of 1-palmitoyl-2-oleoyl-sn-glycero-3-phosphocholine (POPC), 1-palmitoyl-2-oleoyl-sn-glycero-3-phospho-(1′-rac-glycerol) (sodium salt) (POPG), and cholesterol and stained by Texas Red® 1,2-dihexadecanoyl-sn-glycero-3-phosphoethanolamine (triethylammonium salt) (Texas Red DHPE; 0.06 mol% of POPC). The composition was POPC:POPG:cholesterol = 9:1:1 (molar ratio). The liposomes were produced from their lipid mixture film doped with fructose to obtain liposomes of low lamellarity (final concentration of fructose in the liposome dispersion was 1 mM)[34,35], and then, they were introduced and observed in the microfluidic device with 1 mM fructose solution (see details described in Methods). Notably, the peripherals for the microfluidic experiments were developed to be regulated automatically and integratively (MANSIONs). The whole setup was consisting of three programmable syringe pumps, three electric valves, observing apparatus (including microscopy, stage, light source, and camera), and the mixing device and the trapping device (Fig. 1 and Supplementary Fig. 1–6, and Supplementary Discussion).

When the trapped liposomes (the average diameter of the trapped liposomes was 12.1 μm with the coefficient of variation (CV) of 12%) were exposed to a 15 μM uranine/1 mM fructose solution at a flow rate of 40 μL h$^{-1}$, ~80% of the liposomes showed green fluorescence after 15 min in an epi-fluorescence microscopy (EFM) image (Fig. 2a). Line profiles obtained by the scanning-disk confocal fluorescence microscopy (SDCM) image suggested the encapsulation of uranine: the line profile of the membrane (red) had two peaks and was concave between the peaks while that of uranine (cyan) was unimodal and convex (Fig. 2b, c, and Supplementary Fig. 7). Since trapped liposomes in the nests were exposed to considerable flow (Supplementary Movie 1), we focused on the hydrodynamic conditions of the trapped liposomes.

In advance of the further detailed investigations with MANSIONs, first, we prepared liposome dispersions using phospholipids and other chemicals purchased from different companies and confirmed that the dynamics were not caused by unintended contamination or materials of a specific lot number. Next, no degradation of phospholipid molecules used in the liposome dispersion was confirmed for 4 days (Supplementary Fig. 8 and Supplementary Table 1). In addition, simple reference experiments such as vortex and pipetting of the liposome dispersion with uranine did not afford such unpredicted accumulation of uranine into the preformed liposomes (Supplementary Fig. 9).

To more directly verify this unique event of liposomes, we performed the following experiments with MANSIONs. First, we distinguished uranine inside the liposomes from that outside by exposing the trapped liposomes to a low pH solution. The fluorescence intensity of uranine was suppressed at low pH in the used aqueous solution of uranine (Supplementary Fig. 10). We thus changed the outer solution from a 5 μM uranine/1 mM

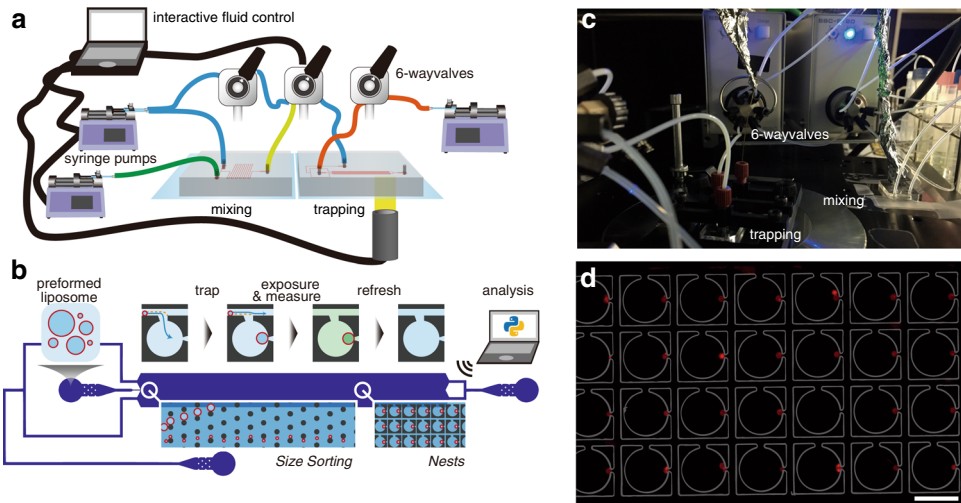

**Fig. 1 Experimental setup of MANSIONs for observing cell-sized liposomes. a, b** Schematic illustrations of the whole system (**a**) and the trapping device (**b**). **c** A picture of the actual setup. **d** Representative epi-fluorescence microscopy image of the trapped liposomes in one field of view. Large circular structures to trap the liposome are overwritten in gray line. Scale bar: 100 µm.

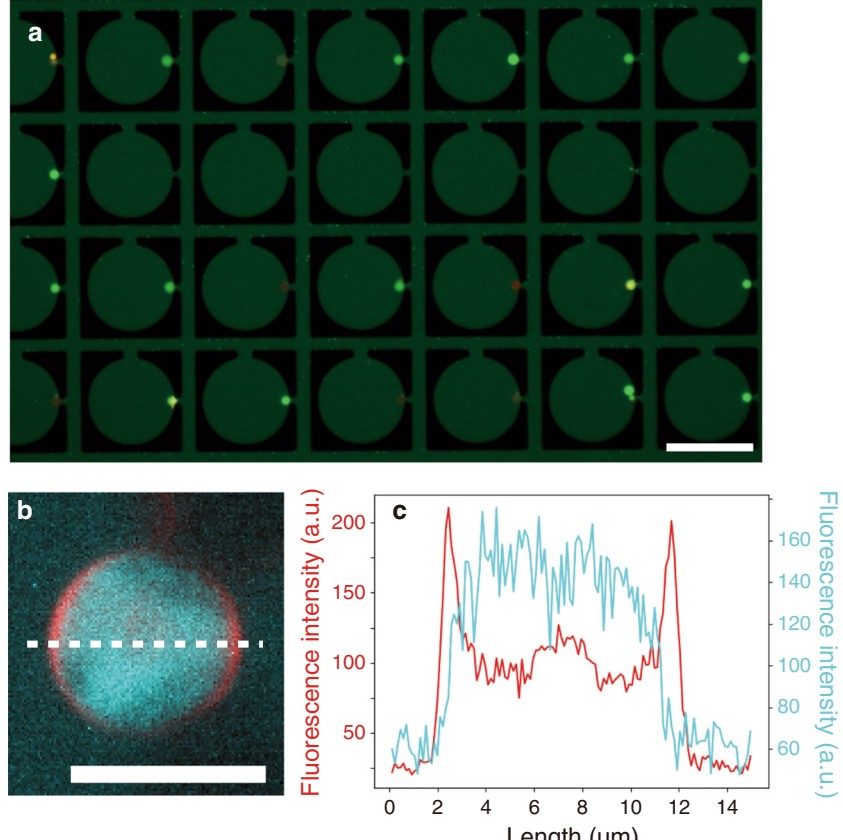

**Fig. 2 Fluorescence microscopy images of liposomes trapped and exposed to uranine solution in the microfluidic device. a** EFM image of trapped liposomes. **b** Representative SDCM image of a trapped liposome. **c** The corresponding line profiles of the liposome in (**b**). Scale bars: 100 µm (**a**), and 10 µm (**b**).

fructose solution (pH 6.42) to a 10 µM HCl/5 µM uranine/1 mM fructose solution (pH 4.68) after trapping the liposomes. As a result, for approximately 30% of trapped liposomes, the green fluorescence intensity (GFI) obtained from the EFM image was maintained at 80% or more of the initial value throughout the exchange of the outer solution ($n = 3$) (Fig. 3a). Note that there were two more different types of the time course of GFI of

liposomes ($GFI_{lipo}$): (i) $GFI_{lipo}$ decayed in the first 5 min during the substitution to the low pH solution, and did not recover during re-substitution to the initial solution ($n = 5$) (Fig. 3b), and (ii) $GFI_{lipo}$ decayed suddenly at a certain time point during the re-substitution process delayed from the first 5 min ($n = 3$) (Fig. 3c). Considering the GFI of the background ($GFI_{BG}$) instantaneously decreased during substitution to the low pH solution, delay or

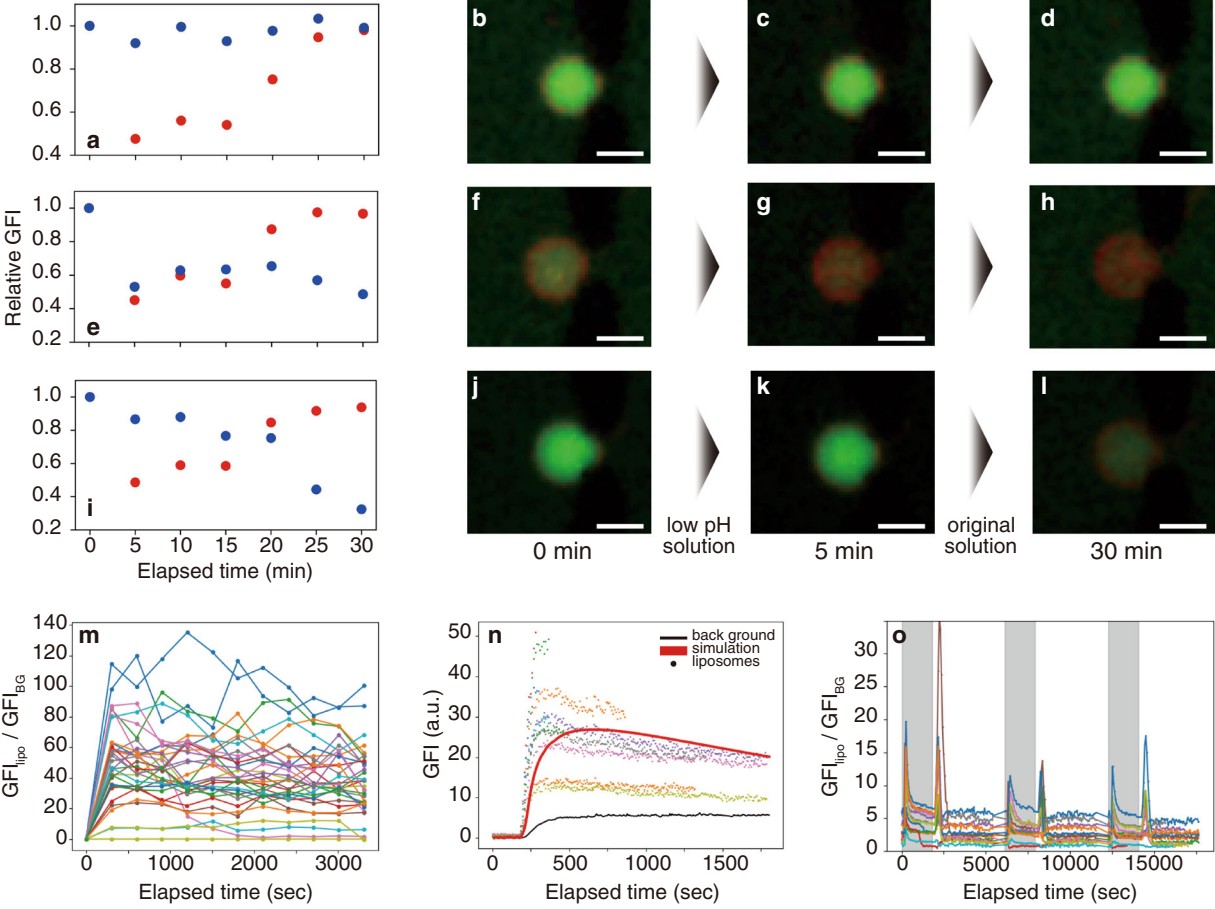

**Fig. 3 Accumulation of uranine in trapped liposomes. a–l** Three types of (**a**, **e**, **i**) time courses of GFI$_{lipo}$ (blue) and GFI$_{BG}$ (red) upon the exposure to low pH solution and following washout with (**b–d**, **f–h**, **j–l**) representative EFM images. The contrast of EFM images was modified for the visual readability. Scale bars: 10 μm. **m** Time course of the ratios of GFI$_{lipo}$ to GFI$_{BG}$ exposed to a 0.5 μM uranine/1 mM fructose solution. **n** Diagram showing the result of the numerical simulation on the time course of GFI$_{lipo}$ (red line) plotted with the experimental results (colored dots, liposomes; black line, background). **o** Time course of the ratio of GFI$_{lipo}$ to GFI$_{BG}$ at a cycle of sequential exposure (5 μM uranine/1 mM fructose; denoted as gray box) and following washout.

total tolerance of fluorescence intensity of uranine against the exchange to the acidic external solution provided a clear proof of the encapsulation of uranine inside liposomes. On the other hand, for about half of the trapped liposomes GFI$_{lipo}$ decreased in the first five minutes. This type of dynamics suggested uranine encapsulation in the shell of the membrane of high permeability of proton or the compiling of uranine on the surface of the liposomes. Considering all the SDCM images of the liposomes exposed to the uranine solution showed common tendency for the line profiles of membrane (bimodal and concave) and uranine (unimodal and convex), these three types of time courses were most likely caused by the variance of the composition of liposomes and hence the permeability[36], and not compiling of uranine on the surface but the encapsulation of uranine occurred at the trapped liposomes.

Second, we examined whether pH gradient between inside and outside of the liposomes caused green fluorescent liposomes upon the uranine exposure or not. Using a buffered uranine solution (pH 7.87) for both the preparation of liposome dispersion and the external uranine solution, GFI$_{lipo}$ of which were larger than GFI$_{BG}$ was observed again when liposomes formed by a buffer solution (pH 7.87) were trapped and exposed to the uranine solution (Supplementary Fig. 11). This result apparently shows the encapsulation and further accumulation of uranine inside of the liposomes, however, the use of pH buffered solution cannot be safe to exclude the hypothetical concentration gradient across

the liposomal membrane. This is because the buffer reagents potentially influence to the phenomena of the accumulation of uranine against the concentration gradient, where might be some unknown dynamics across the membrane.

Therefore, to eliminate the possibility of overvaluation of the uranine concentration inside liposomes caused by the hypothetical concentration gradient between inside and outside of liposomes, we used a more diluted uranine solution (0.5 μM uranine/1 mM fructose) as the outer solution, with the expectation of enhancing the ratio of GFI$_{lipo}$ to GFI$_{BG}$. The pH values of the 0.5 μM uranine/1 mM fructose solution and that of the liposome dispersion ranged from 5.79 to 6.18 and from 6.52 to 7.07 (based on five independent measurements per sample), respectively. Therefore, the overvaluation of uranine concentration inside should be less than a factor of 5.4 (the ratio of GFI at pH = 7.3 to that at pH = 5.7, see Supplementary Fig. 10). GFI$_{lipo}$ at each time point was subtracted by the initial GFI$_{lipo}$ to avoid the effect of the fluorescence crosstalk of the two fluorescent molecules, uranine and Texas Red DHPE. In addition, we repeated the experiments for three times to confirm the reproducibility. As a result as shown in the Fig. 3d, after the 1 h of the exposure, the average ratio of GFI$_{lipo}$ to GFI$_{BG}$ was 37.2, and a gap of the ratio appeared between 17.8 and 6.2, and 90% of liposomes showed the ratio larger than 17.8 ($n = 34$ in total). Importantly, GFI$_{BG}$ (0.5 μM uranine/1 mM fructose) was clearly detected as the meaningful signal compared to that obtained from the 1 mM fructose solution. That is, these

two distributions were both fitted with Gaussian distribution and the mean values were separated from each other by larger than the three times of standard deviation (Supplementary Fig. 12). Thus, even if we assume the hypothetical concentration gradient across the liposomal membrane at the maximum (factor of 5.4), still the additional factors of 3.3 and 6.9 are required to justify the ratio of 17.8 and the average ratio of 37.2. However, judging from the deviation of $GFI_{BG}$ (Supplementary Fig. 12b), the distribution of $GFI_{BG}$ showed that it is not plausible to obtain the additional factor of 3.3 on the ratio of $GFI_{lipo}$ to $GFI_{BG}$ (<0.05% of the trapped liposomes for one measurement). Taking these results into account, we reliably deduce that the concentration of uranine inside liposomes became higher than that of the outer solution.

Third, a numerical simulation was performed (see Supplementary Discussion for the details). The kinetics of photobleaching was estimated from the bulk experiment using the liposome dispersion prepared with uranine (Supplementary Fig. 13). Importantly, in the simulation, the parameters reflecting the permeation of uranine from outside to inside and those from inside to outside the liposome were fitted to be non-equivalent. The numerical simulation moderately reproduced the tendency of the time course of $GFI_{lipo}$ measured during exposure to a 5 μM uranine/1 mM fructose solution (Fig. 3e).

Fourth, we experimentally explored the unequal kinetics of intake and release suggested from the numerical simulation by measuring the time courses of the ratio of $GFI_{lipo}$ to $GFI_{BG}$ at a cycle of exposure (5 μM uranine/1 mM fructose; 30 min) and following washout (1 mM fructose; 60 min). As shown in Fig. 3f, peaked time courses were obtained at the beginning of exposure

and washout respectively. The result indicated the fast incorporation and slow releasing of uranine. The result further proved the repeatability of the cycle of the accumulation and the release of the substances. Taking all of the above-mentioned results into account, we deduce that uranine were accumulated in liposomes due to the non-equivalent kinetics of the intake and release of uranine.

**Effect of hydrodynamic flow and physical contact on uranine accumulation in liposomes.** The accumulation described above is distinguished from the passive diffusion observed in a static condition. To clarify the phenomenological key factor, first, we used another trapping device, in which liposomes are trapped in narrow spaces[35]. As a result, we found that for approximately 75% of the liposomes (n = 166), $GFI_{lipo}$ was larger than $GFI_{BG}$ during exposure to a 5 μM uranine/1 mM fructose solution (Supplementary Fig. 14). Thus, the result suggested that the shape of the trapping structure did not have the critical role on the accumulation. As for the statistical view point of the distribution of $GFI_{lipo}$ shown in the double logarithmic plot, a negative correlation between the $GFI_{lipo}$ and its frequency was observed for the liposomes of high $GFI_{lipo}$ (approximately larger than 40 where $GFI_{BG}$ was 16), while the correlation was not clear for the liposomes of low $GFI_{lipo}$.

Besides, when the liposomes flowed with the 5 μM uranine/1 mM fructose solution in the size-sorting module of the trapping device, the liposomes gradually showed green fluorescence, and the $GFI_{lipo}$ was larger than the $GFI_{BG}$ (Fig. 4a–e, and Supplementary Movie 2).

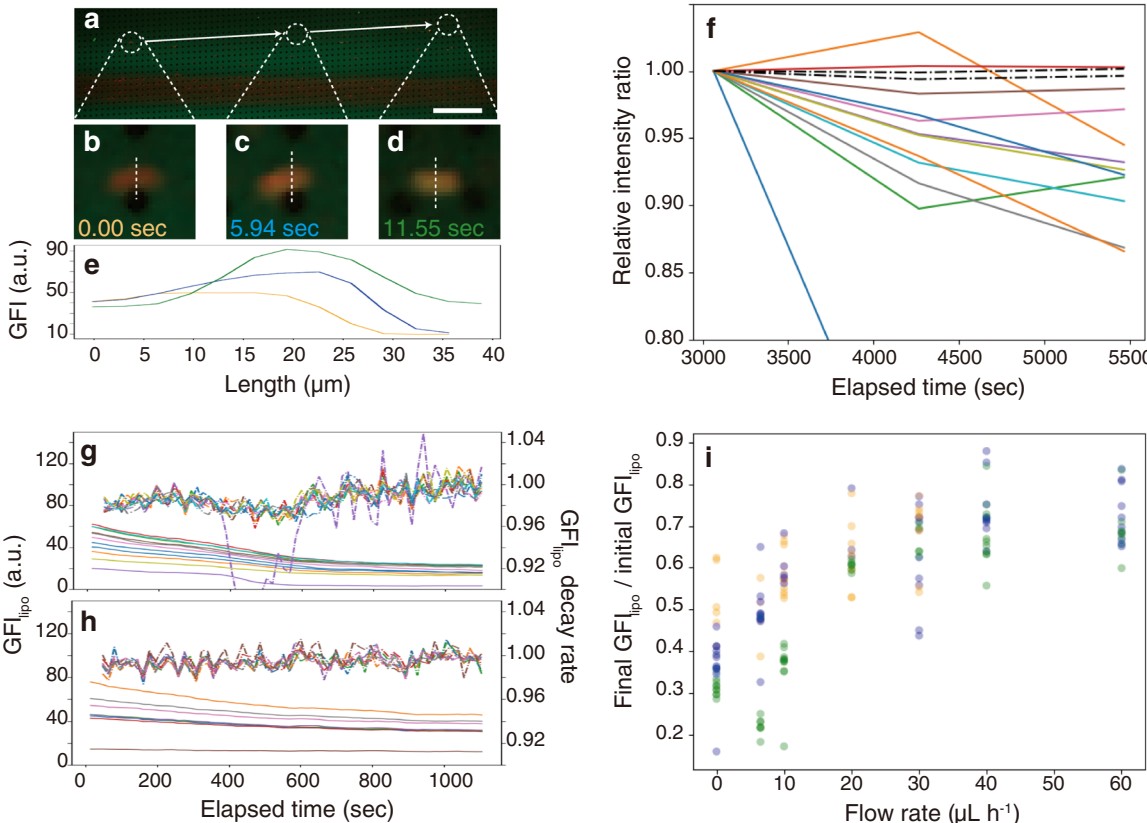

**Fig. 4 Flow rate dependency of the accumulation of uranine in trapped liposomes. a–e** EFM images of liposomes flowing in the size-separating module (**a**) and enlarged images (**b–d**) with corresponding line profiles at 0.00 s (orange), 5.94 s (blue), and 11.54 s (green) (**e**). Scale bar: 500 μm. **f** Time course of the relative ratio of $GFI_{lipo}$ to $GFI_{BG}$ (colored lines) and relative $GFI_{BG}$ normalized by each value at 3000 s. **g–i** Time courses of $GFI_{lipo}$ (solid line) and the decay late of $GFI_{lipo}$ (dashed line) after the flow rate was changed from 40 μL h⁻¹ to 0 μL h⁻¹ (**g**) and 60 μL h⁻¹ (**h**) at 0 s, respectively, and (**i**) dot plot of the ratio of $GFI_{lipo}$ at 1200 s (final $GFI_{lipo}$) to that at 0 s (initial $GFI_{lipo}$). Colors (orange, green, and blue) correspond to three independent measurements.

Since the microposts in the size-sorting module separate particles according to their excluded volume collided to the surface of the microposts[37], intermittent contact to the microposts was imposed on the liposomes. Together, it is strongly suggested that the accumulation was induced by the flow field and physical contact to some surface.

To verify the effect of the flow field on the accumulation, the flow was abruptly stopped after the 30 min of prior exposure to the 5 μM uranine/1 mM fructose solution (for the abrupt stop, see Supplementary Movie 3). In this situation, the concentration of the outer solution was kept constant. Note that to avoid generating photo-bleached uranine, we did not apply the irradiation light to the liposomes before the abrupt stop, and then the measurement was performed with a long interval (interval: 20 min, optical exposure time: 300 ms). As a result, the ratio of the $GFI_{lipo}$ to $GFI_{BG}$ decreased over time (~90% after 60 min) (Fig. 4f). Since $GFI_{BG}$ was almost constant, the decrease of the $GFI_{lipo}$ was attributed to passive diffusion. In other words, a flow field is certainly crucial to accumulate uranine against the concentration gradient.

The effect of the flow field was thus quantitatively examined. We measured the time course of the $GFI_{lipo}$ under flow rates of 0, 6.5, 10, 20, 30, 40, and 60 μL h⁻¹. The flow rate of 0 μL h⁻¹ denoted the abrupt stop of the flow as mentioned above. In this experiment, liposomes were exposed to the 5 μM uranine/1 mM fructose solution at 40 μL h⁻¹ for 30 min in advance so that the outer and inner concentration of uranine were constant and equivalent for all of the flow rates examined. We observed the liposomes over a short interval (interval: 16 s, optical exposure time: 300 ms) to enhance the effect of photo-bleaching. Thus, with faster intake of intact uranine, the time course of the $GFI_{lipo}$ is more likely to become flat. For example, comparing the results of 0 and 60 μL h⁻¹ (Fig. 4g, h) the decay rate (the ratio of $GFI_{lipo}$ to that of the same liposome before 16 s) was maintained near 1.00 for 60 μL h⁻¹, while a sigmoidal transition from ~0.98 to 1.00 was obtained for 0 μL h⁻¹ (for calculation of the decay ratio, original time courses of $GFI_{lipo}$ were smoothened with four consecutive GFI values for each time point). We plotted the ratio of $GFI_{lipo}$ at the final time point (1200 s) to that at the initial time point (0 s) for each flow rate. As a result, this ratio tended to increase with an increase of the flow rate, at least in range from 0 to 20 μL h⁻¹ (Fig. 4i). Namely, the high-flow rate afforded the large difference of kinetics between the intake and release of uranine, resulting in the accumulation.

We also statistically compared the equilibrated distribution of $GFI_{lipo}$ exposed to uranine without irradiating light under different experimental conditions: exposure time $t$ (min), uranine concentration $c$ (μM), and flow rate $v$ (μL h⁻¹); abbreviated as [$t$, $c$, $v$]. The distributions of $GFI_{lipo}$ obtained from three independent experiments for each condition were combined and tested by Mann–Whitney's $U$-test. As a result, the datasets [30, 15, 20] and [15, 15, 40], under which the total amount of uranine was equivalent, were statistically distinguishable in a two-sided test ($p < 0.05$) (Supplementary Fig. 15), suggesting that the flow field affected the accumulation of uranine.

**Permeation and accumulation of fluorescein-tagged ATP.** To examine whether the observed accumulation is extended to other small molecules, especially those that are biologically relevant, we exposed the trapped liposomes to an aqueous solution of an ATP analogue, fluorescein-12-adenosine triphosphate (FL-ATP). The concentration and flow rate of the outer solution were fixed at 15 μM FL-ATP/1 mM fructose and 40 μL h⁻¹, respectively, and EFM images were taken every 5 min throughout the following experiments.

When the liposomes were exposed to the FL-ATP solution, 40% of measured liposomes showed an increase in the $GFI_{lipo}$ that was approximately three times higher than the $GFI_{BG}$ in the first 5 min, and the fluorescence inside was detectable even after the 4 h of washout by 1 mM fructose solution (Fig. 5a, b). With the same experimental procedure but using fluorescein instead of FL-ATP as a reference experiment, $GFI_{lipo}$ also became higher than $GFI_{BG}$ during exposure to the fluorescein solution, but after the 30 min of washout, $GFI_{lipo}$ was almost the same as the initial value before exposure (Supplementary Fig. 16). Therefore, the intake and release of FL-ATP was remarkably more imbalanced than fluorescein probably because of its ATP moiety.

Interestingly, we found a non-monotonic time course of the $GFI_{lipo}$ during exposure to the FL-ATP solution (15 min) and following washout (Fig. 5c). $GFI_{lipo}$ decreased rapidly during the first 5–10 min, and then gradually recovered during the washout. This characteristic time course can be explained by the small pKa of ATP (the first pKa is less than 1.0)[38] and resulting suppression of the fluorescence emission of fluorescein under a low pH condition (Supplementary Fig. 17). Namely, the fluorescence from FL-ATP was suppressed along with an increase of the concentration due to its own acidity.

This non-monotonic time course of $GFI_{lipo}$ is assigned to a proof of higher concentration of FL-ATP inside than that of the outer solution. First, the recovery of fluorescence intensity of FL-ATP during the washout (30–60 min) suggested the continuous encapsulation of FL-ATP during 5–20 min. Second, the strength of suppression of fluorescence from FL-ATP was positively related to the amount of FL-ATP. Third, at a couple of liposomes, $GFI_{lipo}$ was higher than $GFI_{BG}$ at 5 min although at 10 min the

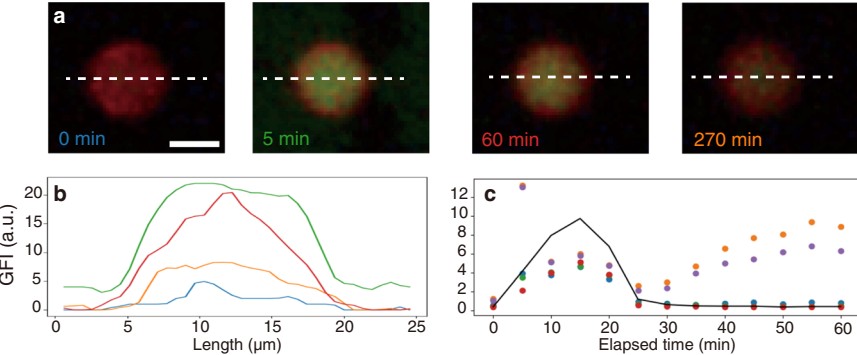

**Fig. 5 Permeation of FL-ATP into liposomes. a** Representative EFM images of a liposome exposed to FL-ATP. The contrast was modified for the visual readability. Scale bar: 10 μm. **b** Line profiles of the liposome at 0 (blue), 5 (green), 60 (red), and 270 (orange) min (indicated as white dashed lines in (**a**)). **c** Whole time course of $GFI_{lipo}$ (colored dots) and $GFI_{BG}$ (black line).

fluorescence intensity of FL-ATP inside the liposome was more strongly suppressed than the outer solution. A stoichiometric estimation demonstrated that the amount of FL-ATP supplied to one liposome is enough to suppress the fluorescence from FL-ATP due to its acidity (see Supplementary Discussion for the detail). Therefore, the concentration of FL-ATP inside liposomes upon the FL-ATP exposure was higher than that of the outer FL-ATP solution.

**Potential mechanism for the enrichment of molecules and ions**. Permeation of molecules and ions across the lipid bilayer membrane in a static state is primarily governed by a partitioning process rather than pore formation in the bilayer membrane of POPC[39], which was the main component of the liposome in our experiment. Researchers previously reported the passive permeation of lipophilic molecules[40–42], and clarified that the passive permeation and adsorption of ions to the membrane is enhanced by shear force[43,44] and by hydrostatic pressure[45], respectively. Therefore, it is reliable that passive diffusion of the charged fluorescent molecules was also accelerated in MANSIONs since syringe pumps imposed steady shear and pressure to trapped liposomes. However, the results presented here shows the non-equivalent permeation kinetics of small molecules and ions between intake and release across the liposomal membrane, which requires another mechanism.

First, we investigated the effect of temperature. Under high temperature, it was expected that membrane fluidity and permeability would increase for both intake and release of uranine, and as a result, the difference of these two kinetics would be reduced, and the accumulation would cease. However, even when we increased the temperature (~40 °C) from the room temperature (25 °C) using instant heat pads placed near the device, the accumulation of uranine was observed (Supplementary Fig. 18).

Second, we tested the effect of membrane constituents other than POPC, that is, fructose, cholesterol, and POPG. In case using liposomes prepared in the absence of fructose or cholesterol, the trapped liposomes with low lamellarity having the $GFI_{lipo}$ larger than $GFI_{BG}$ were observed during the uranine exposure (Supplementary Fig. 19). However, liposomes prepared without POPG did not trapped in the device even after 12 h of introduction of the liposome dispersion because of their smaller diameters of liposomes. When we used newly designed and fabricated device to trap smaller liposomes, the number of liposomes showing the higher value of $GFI_{lipo}$ than that of $GFI_{BG}$ (Supplementary Fig. 20a) were negligible (1 in 15). Note that even in this trapping device, for 66% of trapped liposomes prepared with POPG, $GFI_{lipo}$ was larger than $GFI_{BG}$ ($n = 45$; Supplementary Fig. 20b).

Significantly, temperature and cholesterol are known to affect the membrane fluidity and hence the permeability[46,47], while POPG does not influence at least the water permeability[35]. Thus, we focused on the anionic charge of POPG for the unequal transportation of substances across the phospholipid bilayer membrane. In fact, the zeta potentials of the liposome dispersions prepared with and without POPG were $-74.9 \pm 1.5$ mV and $-35.7 \pm 0.27$ mV respectively ($n = 3$ for each condition) (Supplementary Fig. 21). This result indicates that the liposomes accumulating the small molecules upon the external flow of their solutions had strongly negative surface charge.

Besides, if the non-equivalent permeation is accompanied by pore formations, then the permeation of water-soluble small molecules would be equally enhanced. Thus, we examined 2,5-bis (methylsulfonyl)-1,4-phenylenediamine (BMeS-p-A), which has a smaller molecular weight than uranine and is known as a pH-independent fluorescent molecule[48]. However, intake of BMeS-p-A

into the trapped liposomes was not detected even when a 15 μM BMeS-p-A/1 mM fructose solution was introduced to the trapped liposomes for 60 min under 40 μL h$^{-1}$ (Supplementary Fig. 22). Therefore, partitioning should be the dominant mechanism rather than pore-formation.

Interpreting all the results together, currently we can postulate that the mechanism of the hydrodynamic accumulation of substances across the liposomal membrane against the concentration gradient is an asymmetric distribution of POPG between inner and outer leaflets. When the liposome was imposed to the surface of the device upon the external flow, the contact caused transient negative curvature and some extent of membrane perturbation, which might be a critical process to bias the distribution of POPG. As a result, the surface charge of the liposome might be imbalanced between inside and outside of the liposomal membrane. The permeation of the small molecules and ions through partitioning process was influenced by the surface charge of the membrane, and then, the entrapped substances were more strongly inhibited to permeate outward. Further investigations and elucidation are desired to fully understand this unique property of the liposomes.

**Significance of the current findings**. Here we demonstrated that small molecules and ions were encapsulated and rather accumulated into liposomes even against the concentration gradient under microfluidic environment. With the machine assisted measurement based on the direct observation of cell-sized liposomes (MANSIONs), we quantitatively and statistically clarified that the accumulation even against the concentration gradient was caused only by an external flow and physical contact with a substrate. Since the flow field is easy to be expected even in the prebiotic era, and our system do not contain any sophisticated proteins, the phenomena we found here could be implemented to wide range of possible protocells. On the other hand, it is noteworthy that the hydrodynamic accumulation was triggered by the flow field, but strongly depended on the chemical composition of the liposomes, and not the all molecule was accumulated as shown by BMeS-p-A. Although full elucidation of this puzzling permeation process including the mechanism would require further researches, the observed molecular dependencies can provide a clue to consider a molecular scenario for the continuous development of protocell in the early earth by suggesting a promising combination of molecular species and compartment enabling to earn and retain substances repeatedly through the phospholipid membrane. It was also worth emphasizing that FL-ATP is an analogue of a chemical energy source to drive enzymatic reactions and other biological reaction networks. Thus, the hydrodynamic accumulation even against the concentration gradient presented here could be a practical methodology to provide energy source to liposome-based cell models which would lead remarkable progress of synthetic approach and the investigation of the origin of life.

## Methods

**Chemicals**. 1-Palmitoyl-2-oleoyl-*sn*-glycero-3-phosphocholine (POPC) and 1-palmitoyl-2-oleoyl-*sn*-glycero-3-phospho-(1′-rac-glycerol) (sodium salt) (POPG) were purchased from Avanti Polar Lipids (Alabaster, AL, USA) and NOF Corporation (Tokyo, Japan). Cholesterol was provided by Avanti Polar Lipids. Texas Red® 1,2-dihexadecanoyl-*sn*-glycero-3-phosphoethanolamine (triethylammonium salt) (Texas Red DHPE) was supplied by Thermo Fisher Scientific, Inc. (Waltham, MA, USA). Fructose, uranine, fluorescein, sodium hydroxide, hydrochloric acid, methanol, chloroform, and pH standard solutions (phthalate pH standard solution, phosphate pH standard solution, phosphate pH standard equimolal solution, and tetraborate pH standard solution) were purchased from FUJIFILM Wako Pure Chemical Corporation (Osaka, Japan). 2,5-Bis(methylsulfonyl)-1,4-phenylenediamine (BMeS-p-A) and a solution (1 mM) of fluorescein-12-adenosine triphosphate (FL-ATP) with 10 mM Tris-HCl, pH 7.6, 1 mM EDTA were supplied by Tokyo Chemical Industry Corporation (Tokyo, Japan) and PerkinElmer, Inc. (Waltham, MA, USA), respectively. Fluorescent microbeads (Fluoresbrite® YG Microspheres, Calibration Grade 1.00 μm) were

provided by Polysciences, Inc. (Warrington, PA, USA). All materials were used without further purification.

**General procedure to prepare the liposome dispersion**. Stock solutions of POPC (40 mM in chloroform; 50 μL), POPG (4.44 mM in chloroform; 50 μL), cholesterol (4.44 mM in chloroform; 50 μL), Texas Red DHPE (14.5 μM in chloroform; 83 μL), and fructose (40 mM in methanol; 50 μL) were mixed in a glass vial. When we used liposomes without fructose, cholesterol, or POPG, the addition of corresponding stock solutions was omitted. The organic solvents were removed with a rotary evaporator (EYELA, N1110V) and then dried under reduced pressure at 40 °C for 1 h to prepare a dry lipid film. The film was gently agitated with 2 mL of Milli-Q water or 1 mM Tris-HCl buffer (pH 7.87) in a thermostatic incubator (WAKENYAKU, MODEL 2290) under 40 °C for 1 h. Then, the liposome dispersion was placed into the thermostatic incubator under 26 °C for more than 10 h. The liposome dispersion used for the microfluidic experiments was diluted threefold by 1 mM fructose solution and incubated at 26 °C for more than 1 h, and then the diluted dispersion was filtered by a nylon mesh filter (Merck Millipore Ltd., NY2002500) with a pore size of 20 μm four times to avoid the clogging in the microchannel, the height of which is ~16 μm. Filtrated liposome dispersion was placed in the thermostatic incubator under 26 °C for more than 3 h.

**Image acquisition**. Microscopic images were taken using an inverted microscope (Olympus, IX71) equipped with a cooled charge-coupled device camera (Olympus, DP72) through commercially available software (Molecular Devices, MetaMorph). The control of the shutter unit (Ludl Electronic Products Ltd., Mac 6000) and the electric stage (Sigma koki) was also achieved through MetaMorph. We wrote an ad-hoc regulating code of MetaMorph, called journal. For the EFM image, a dual band excitation filter (excitation: 490–505 nm, 560–580 nm; emission: 515–545 nm, 600–650 nm) was typically used. The microscope also equipped an apparatus for the spinning-disk confocal fluorescence microscopy (SDCM) observation: a confocal scanning unit (Yokogawa Electric, CSU22) and a scientific complementary metal-oxide semiconductor camera (Andor, Zyla) under control of an image capture software (Andor, iQ3).

**Microfluidic fabrication**. The microfluidic devices used in this study were designed with a commercially available software (Tenner Inc., L-Edit). The device design was transferred to a photomask using a mask-less lithography system (Nano System Solutions, D-Light DLS-50). Ultraviolet light was irradiated to SU8-10 (MicroChem Co.) plated on a silicon wafer through the photomask to obtain a master mold. Polydimethylsiloxane (PDMS)-based resin (Dow Corning, SilPot 184) was poured onto the master mold with a curing agent (Silpot 184 CAT), and then heated on a heat plate (75 °C) for more than 2 h. The PDMS block and glass slide (Matsunami Glass Industries, 0.25–0.35 mm in thickness) were quickly bonded after O$_2$ plasma treatment (SAMCO, FA-1). The microfluidic device was then heated on the heat plate (75 °C) for more than 1 h.

**General procedure for the initial set-up of the microfluidic devices**. For the trapping device, 1 mM fructose solution was introduced into the device from the outlet for the outer solution at 3000 μL h$^{-1}$ through the tube for the outer solution. When the solution reached the inlets and a droplet was formed in the vicinity of the inlets, the tube was re-connected to the inlet for the outer solution, and a guide tube was connected to the outlet for the drain. Subsequently, a tube for liposome dispersion was connected to the device with flowing liposome dispersion to prevent trapping air in the device. To remove the small bubbles inside the device, the tube of the liposome dispersion was dammed back by changing the valve position to impose inner pressure (see Supplementary Fig. 1a). After the device was fully filled by the aqueous solution, the flow rate of the outer solution was set to 1000 μL h$^{-1}$, and the tube of the liposome dispersion was connected to the trapping device by valve manipulation as described above. The flow rate was slowed down to 300 μL h$^{-1}$ (1 min), 100 μL h$^{-1}$ (1 min), and finally 40 μL h$^{-1}$. The typical flow rate of the liposome dispersion was 6.5 μL h$^{-1}$. After this first set-up, all of the procedures were operated as programmed in advance (see the later section). The same procedure was applied for the mixing device.

**Vortex and pipetting of the liposome dispersion with uranine solution**. The liposome dispersion was prepared as explained above. The dispersion was diluted by 10-fold to afford a 15 μM uranine/1 mM fructose solution. The dispersion was then vortexed at 500 rounds per minute for 60 min or pipetted vigorously with a pipet for more than 5 min. The sample was placed onto a 25-μL specimen with two cover glass slips (thickness ~280 μm) and observed with EFM and SDCM. Several images were taken for each condition and the incorporation of uranine was evaluated by the line profile over the liposomes.

**Observation of flow in an occupied nest**. To avoid aggregation of microbeads to the surface of the device due to ruptured lipids on the surface, the green fluorescent microbeads dispersion (1 μL) was diluted to 5 mL by 1 mM fructose solution to afford a microbeads dispersion.

The liposome dispersion was prepared, and the trapping device was set as explained above. After a moderate number of liposomes were trapped in the nests, the valve position was changed to shut off the flow of the liposome dispersion. After 5 min, the valve position was changed again to exchange the outer solution from a 1 mM fructose solution to the microbeads dispersion. The exposure time was 300 ms and the flow rate was 40 μL h$^{-1}$. Because of the low density of microbeads and fast flow, we observed the stroboscopic images for the trace of microbeads instead of the streamline inside the occupied nest.

**Nuclear magnetic resonance measurement of the liposome dispersion**. The liposome dispersion was prepared as explained above. A part of the prepared liposome dispersion (~500 μL) was placed in a microtube and rapidly frozen by liquid nitrogen. The tube was then placed in an insulated glass container and freeze-dried for more than 12 h. The dried lipid mixture was dissolved into deuterated methanol (methano-$d_4$) and the $^1$H NMR spectra were obtained by AVANCE 500 (Bruker). The freeze-drying of the liposome dispersion was conducted immediately after, and then at 1 and 4 days after the preparation of the liposome dispersion.

**Typical procedure for data acquisition in MANSIONs**. At the beginning of the in-house developed Python program, the areas of the trapping microstructure (termed nests) were detected through Hough transformation of the binarized bright-field image of the trapping region taken in advance. In the current trapping device, there are 28 nests in one field of view. When the fluorescence image was taken, the red channel of the image was separated and decomposed into 28 small binarized images based on the location of each nest. We adopted Otsu's method for the binarization. If only one object with moderate size and circularity was in the decomposed image, the object was judged as a liposome. The binarized image was also used as a mask of the original image to measure the value of the liposomes. Size, centroid, and averaged intensity of each channel (blue, green, and red; abbreviated as BFI, GFI, and RFI, respectively) were recorded over time. Time zero was determined as the time when the Python program was started. When certain criteria were achieved during the measurement, grayscale images were output to a folder to regulate the microscope. The journal script of MetaMorph searched the folder twice per second to read regulating images and operated as programmed. Computer-controllable pumps (Harvard Apparatus, Pump 11 Pico Plus Elite and YMC, YSP-202) and valves (Senshu Scientific Co., Ltd, SSC-9720) were also regulated based on the image analysis by the Python program. Note that the valve with transistor-transistor-logic (TTL) regulation was controlled through a digital I/O terminal (Contec Co. Ltd., DIO-0808LY-USB). We named this automated observation platform for MANSIONs. Note that SDCM image of liposomes entrapped in the microfluidic device can be taken under the same experimental setup by simply pausing MANSIONs, although the measurement requires manual regulation of the focal plane.

**Investigation of the relationship between the fluorescence intensity of uranine and pH**. A commercially available pH meter (HORIBA Advanced Techno, Co., Ltd., Model: D-51) equipped with an ISFET pH electrode (HORIBA Advanced Techno, Co., Ltd., Model: 0040-10D) was used for the pH measurement. The pH meter was calibrated by three points with standard solutions: phthalate pH standard solution (pH 4.01), phosphate pH standard solution (pH 6.98), and tetraborate pH standard solution (pH 9.18). Specimens (~1 mL) were placed onto a glass slide and measured as a droplet on the solid surface according to the manufacturer manual. The same procedure was adopted throughout the measurement of pH values in the latter part. We prepared 14 types of aqueous solutions of different pH values from the standard pH solution with sodium hydroxide and hydrochloric acid (pH 4.41, 4.82, 5.13, 5.70, 6.03, 6.51, 6.86, 7.26, 7.67, 8.21, 8.82, 9.18, 9.75, and 11.22). Uranine solution was added to obtain a uranine concentration of 5 μM. The pH value of each sample was measured again and put onto a 25-μL specimen with two cover glass slips (thickness ~280 μm). Fluorescence images were taken with an exposure time of 25 ms. The images were cropped for analysis of common regions of interest to remove contaminants, and the GFI was measured.

**Microfluidic observation with another observation device trapping liposomes in narrow nests**. The microfluidic device was fabricated as explained above according to the design described in our previous report[35]. The flow rate of the outer solution and liposome dispersion was 40 and 6.5 μL h$^{-1}$, respectively. After a moderate number of liposomes were trapped, the valve was changed to shut off the flow from the pump introducing the liposome dispersion (see Supplementary Fig. 1a). The outer solution was substituted to the 5 μM uranine/1 mM fructose solution at a flow rate of 40 μL h$^{-1}$, and the fluorescence image was taken after 30 min of exposure to uranine. The image was manually analyzed by imageJ for all of the liposomes in the image: the contour was manually fit with the red channel of the image and GFI was measured for the region of interest.

**Microfluidic observation with newly designed and fabricated device for trapping smaller liposomes prepared without POPG**. The size-sorting module at the front of the nests was designed to separate liposomes the diameter of which

were larger than 5.0 μm. The nest structure was basically same to the one described in Supplementary Fig. 2, but the shape of the exit channel was the rectangle the width and the height of which were 3 and 8 μm respectively. The microfluidic device was fabricated as explained above. After the bonding, trimethoxy (1H,1H,2H,2H-heptadecafluorodecyl)silane/methanol solution (10 wt%, with an aliquot of 6 M hydrochloric acid) was introduced into the device for 60 min. Then the device was washed with excess amount of methanol and Milli-Q water. Microfluidic observation was proceeded as explained above.

**Observation of flow upon the abrupt stop of the flow**. The green fluorescent microbeads dispersion (25 μL) was diluted to 5 mL by Milli-Q water. The dispersion was introduced into a microfluidic device, and a movie was captured with the optical exposure time of 300 ms.

**Statistical analysis for uranine concentration inside trapped liposomes**. After the programmed experimental procedure was finished, the recorded $GFI_{lipo}$ was converted to a pseudo-concentration ($C_{calcd}$) to avoid the influence from the deterioration of the mercury lamp over time using the following formula:

$$C_{calcd} = GFI_{lipo} \times \frac{C}{GFI_{BG}} \quad (1)$$

where $C$ is the actual concentration of the uranine solution and $GFI_{BG}$ denotes the GFI of the background. The distributions of the pseudo-concentration under each experimental condition were statistically compared by a two-sided test.

Owing to the uncertainty of the noise distribution, we chose the non-parametric Mann–Whitney $U$-test for the statistical analysis, in which the data sets ($k = 1, 2$) were ranked ($r_{kn}$) by their calculated concentration. The statistic index ($U_k$) was calculated based on the number of samples ($N_k$) and the sum of the rank for each data set $\left(R_k = \sum_{n=1}^{N_k} r_{kn}\right)$. In our experiments, the number of samples was sufficient to calculate the $p$-value as the probability of $Z$ distributed as a normal distribution:

$$U_k = N_1 N_2 + \frac{N_k(N_k+1)}{2} - R_k \quad (2)$$

$$Z_k = \frac{\left|U_k - \frac{N_1 N_2}{2}\right|}{\sqrt{\frac{N_1 N_2 (N_1 + N_2 + 1)}{12}}} \quad (3)$$

**Microfluidic observation under high temperature**. Basic experimental procedures for the microfluidic observation with MANSIONs were performed as explained above. Five instant heat pads (Lotte, "HOKARON mini") were put onto the microscopic stage to surround the trapping device. The opaque curtain surrounding the entire apparatus of MANSIONs and an electronic heater were enclosed as a heat source (150 °C). The temperature of the atmosphere in the vicinity of the device and the water temperature from the outlet of the device were measured by an alcohol thermometer.

**Measurement of the zeta-potential of liposome dispersion**. The liposome dispersions prepared with and without POPG were sonicated 20 min and stored under 26 °C for 12 h. The liposome dispersions were then introduced into a quartz-made flow cell (EZ1-870, Otsuka Electronics Co., Ltd., Japan) and measured by ELSZ-1000ZXCK (Otsuka Electronics Co., Ltd., Japan).

## Data availability
The authors declare that the data supporting the findings of this study are available within the paper and its supplementary information files.

## Code availability
Original codes for MANSIONs are deposited to https://park.itc.u-tokyo.ac.jp/toyota_lab/.

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

## Acknowledgements

We appreciate Prof. Yuichi Wakamoto (The University of Tokyo) for his helpful discussions. We thank to Hirofumi Yoshida (Institute of Industrial Science, The University of Tokyo) and Reiko Okura (The University of Tokyo), for their advices and supports for the fabrication of microfluidic device. We acknowledge to Yiting Zhang (Chiba University) for his supports for the zeta-potential measurement. We thank Daisuke Watanabe, Akinori Izumi, and Haruna Nakagawa (The University of Tokyo) for their fruitful group works on the part of preliminary experiments. Part of this study was supported by Grant-in-Aid for JSPS Research Fellow (to H.S.; Grant number JP18J22004), The ANRI fellowship (to H.S.), Scientific Research (B) (to T.T.; Grant number JP16H04032), and Platform for Dynamic Approaches to Living System from The Ministry of Education, Culture, Sports, Science and Technology, Japan.

## Author contributions

H.S., T.O., S.T., and T.T. conceived and designed the experiments. H.S. performed the experiments, analyzed data, and wrote the original manuscript. All authors discussed the results and edited the manuscript with convincement.

## Competing interests

The authors declare no competing interests.
