## [Peer Review File · Communications Chemistry]

Reviewers' comments:

Reviewer #1 (Remarks to the Author):

See attachment

Reviewer #2 (Remarks to the Author):

This manuscript reports the rapid enrichment of small molecules and ions in a liposome upon filter extrusion occurring infrequently, and under a microfluidic environment against the concentration gradient with high yield. By using microfluidic devices manipulated automatically by a machine, termed MANSIONS, based on the direct observation of cell-sized liposomes, the authors clarified that the condensation was induced by the flow field and physical contact to some surface. The paper is nicely written and illustrated. I found the study to be informative.

My suggestions for revision include in order:

Q1: The condensation has been well-known already as shown in the following paper:
J Mol Evol. 2014 Dec; 79(5-6):179-92. doi: 10.1007/s00239-014-9655-7. Epub 2014 Nov 22.
Spontaneous encapsulation and concentration of biological macromolecules in liposomes: an intriguing phenomenon and its relevance in origins of life.
So what are the differences and innovations of this paper?

Q2: The infrequent condensation under the filter extrusion and the frequent condensation discovered in the micro-chamber may not be driven by the same mechanism. Please show the flow rate dependency of the infrequent condensation.

Q3: What is the main reason for non-equivalent kinetics of the intake and release of small molecules?

Q4: This paper should be improved by including direct experimental evidence to support the mechanism for the condensation speculated as follow:
"The outer leaflet of the trapped liposome is disturbed by contacting with a substrate, while the inner leaflet is kept rather intact. Thus, the energetic barrier for the partitioning could be asymmetric between the inside and outside, resulting in hydrodynamic condensation."

Reviewer #3 (Remarks to the Author):

The paper "Hydrodynamic condensation of small molecules and ions into cell-sized liposomes" focused on a very important problem: the emergent properties of biological cells, and the spontaneous mechanisms that can have played a key role in the origin of the first biological cells. In this paper the Authors describe an abiotic mechanisms operated by microfluidic forces, which is able to generate condensation of small molecules in cell-sized liposomes. This is an original finding, of high relevance in the field of origin of life, and of clear interest for all the researchers working on self-organizing systems. The methodology and the approach are both correct and valid, and the presentation is clear. The only weak point in this paper, in my opinion, is the statistical discussion of the data, which is very limited. Since the Authors claim that the condensation is an infrequent event, one would expect also a discussion about the statistical distribution of the filled liposomes vs empty liposomes. Is this a power-law distribution as it happens for another spontaneous liposome filling, like the "supercrowding effect"? (about this, see: Luisi, Pier Luigi, et al. "Spontaneous protein crowding in liposomes: A new vista for the origin of cellular metabolism." ChemBioChem 11.14 (2010): 1989-1992.). I suggest the Authors to integrate the paper with a

statistical discussion and with a comparison between their results about the condensation of small molecules and the supercrowding of high-molecular weight proteins described in the above cited paper.

In this manuscript, the authors describe the effect of solute accumulation inside of 1:1:1 POPC/POPG/cholesterol micro-sized liposomes in presence of fructose

Liposomes were prepared by the swelling method and then they were extruded having being forced to pass through polycarbonate filters of 20 or 5 micro meters (μm) pores.

The solute accumulation has been observed both during the extrusion procedure (5 μm extruded liposomes in presence of 15 μM uranine/1 mM fructose solution) but also in dynamic conditions under a microfluidic environment by exposing trapped 20 μm extruded liposomes to a flowing solution.

The authors try to rationalize the observed effect by a possible mechanism based on the interaction of liposome with a solid surface. If I have well understood, the liposome-solid surface interaction might perturb the outer layer respect to inner one, so inducing an asymmetry in the solute partitioning.

Although the effect described in this paper is quite interesting, I do not think the manuscript is worthy to be published in the present form.

Firstly, this effect is not so new to justify a letter, moreover the mole of data is so large to be better adapted in an ordinary article allowing the reader to follow better the subject if the manuscript is written in a less concise way.

First observations on this effect have been reported by the Luisi's group 9 years ago calling it the crowding effect. See for instance:

Luisi et al. 2010, *ChemBioChem*, Spontaneous protein crowding in liposomes: A new vista for the origin of cellular metabolism.

or

Souza et al. 2014, *Journal of Molecular Evolution*, Spontaneous encapsulation and concentration of biological macromolecules in liposomes: an intriguing phenomenon and its relevance in origins of life.

and in the same field a more recent study can be considered

Fanti et al., 2018 *Integrative Biology*: Do protocells preferentially retain macromolecular solutes upon division/fragmentation? A study based on the extrusion of POPC giant vesicles

see also papers cited there for other examples. All these articles are unfortunately missed among the references of the present manuscript. The authors should at least mention them, although it would be very important to discuss the differences from and the novelty of the present work.

Moreover, the phenomenon of the increase of membrane ion permeation by shear stress or by hydrostatic pressure is well known as reported in ref 36-38. Therefore the only novelty here, appear to be the application to this effect of the MANSION technology, already presented elsewhere refs 27

The phenomenon described here is mainly due to the far from equilibrium conditions which liposome are formed in or they are exposed to. In these cases, the real fate the lipid compartment follows, determines also the final state obtained. Therefore, compare the solute accumulation in liposome formed by the extrusion with that obtained by exposing lipid vesicles to a solute flow, it is for me conceptually wrong. In the first case, in fact, liposomes undergo a strong mechanical stress, in the second case, instead, just a hydro-dynamical perturbation. In my opinion, it is not really based on solid arguments to use this comparison, along with the absence of any observed accumulation by pipetting or vortexing liposomes in a solute solution, in order to state that this effect is driven by the interaction with a solid surface. (Rows 34-35)

In fact, when dimensionally poly-dispersed giant vesicles, prepared by the swelling method, are forced to pass 10 times through a filter of 5um pores in presence of an external solute, the lipid membrane is broken and resealed many times. This process is completely different from exposing a preformed 20um extruded liposome to a solute flow. To assume that the mechanism of solute accumulation is comparable and based on the surface interactions needs more solid argumentations.

Moreover, in the fig 2 extended Data liposomes are reported that have been vortexed and pipetted. Some of them in the enlarged image exhibit an inside region darker than the background or with a similar intensity that should suggest a solute accumulation or a similar concentration. Could you report on which compartments of the enlarged image the line profile have been determined by you or, alternatively, could you determine the line profile on those I have circled?

Moreover, why do you extruded on a different sizes pore filter the liposomes for the hydrodynamic pressure experiments, 20um instead of 5um? Please explain somewhere, since this make further incomparable these two procedures.

What is the exact composition of your liposomes? In the preparation procedure you stated that you swell a lipid film by approximately this composition 1:1:1:10 POPC/POPG/cholesterol/Fructose. Instead, in the main text (row 230) you declare that POPG is the main component of the membrane, while in the supplementary material (note for the quality of phospholipids) you show the NMR only of the POPC like the main component of the membrane. That's a bit confusing!! It is difficult to rationalize an effect without knowing which kind of compartments we are speaking on!!

Assuming that the membrane composition is 1:1:1 POPC/POPG/cholesterol, The different behaviour of the membrane transport the authors observed when liposomes are exposed to a solute flow can be rationalize as due to the negatively charged membrane that inhibits the passive diffusion of the negatively charged fluorescent molecules after the internal accumulation due to hydrostatic pressure.

It is not clear to me the role of fructose in the liposome preparation, the author should comment why this compound is necessary and why it is dried along with lipids in the film and not added to the water solution for the swelling, as normally it is done.

In my opinion, the authors should completely re-think this work focusing it only on the study of the accumulation of solute by hydrodynamic pressure by using the MANSION apparatus.

Some Minor Concerns.

I suggest to the authors to use the word 'accumulation' instead of 'condensation', since condensation is used in physical chemistry for describe the matter transformation from the liquid to the solid state with the formation of stronger bonds among molecules, instead accumulation is used in biological context to describe amount increase of different substrates in cells.

Please somewhere at the beginning declare the composition of the membrane of your liposomes that could make the difference in the behaviour they exhibit.

Row 33- the meaning of sentence is not quite clear to me. What do the authors mean with “to assemble substances”? Maybe they mean: “to concentrate” or “to gather” substances.

Rows 37-39, please insert a reference to justify the statement

Row 45. Add references for MANSION

Row 56: Please explain while the membrane is fluorescent

Row 80: I have not well understood how you collected the liposomes for the images in Fig 2e,f and extended data fig. 3. In fact, the hydrodynamic pressure experiments are done with the MANSIONs equipment shown in fig. 1 where EFM is used.

Row 87 Fig. 2 caption, Please add the pore size of the filter for both the cases in order to understand the different size of liposomes

Row 109: Maybe ‘slower’ is better than ‘smaller’ speaking about the time of proton release

Row 234: The composition of the lipid film you form and swell by Millipore water before the extrusion has a composition 1:1:1:10 POPC/POPG/cholesterol/fructose are you sure that the POPG is the main component of your liposomes.

Extended Data

Fig. 2. Please add the size of the bar.

Point-by-point response sheet

Reviewer #1

Comment #1- Although the effect described in this paper is quite interesting, I do not think the manuscript is worthy to be published in the present form. Firstly, this effect is not so new to justify a letter, moreover the mole of data is so large to be better adapted in an ordinary article allowing the reader to follow better the subject if the manuscript is written in a less concise way. First observations on this effect have been reported by the Luisi's group 9 years ago calling it the crowding effect. See for instance:

Luisi et al. 2010, ChemBioChem, Spontaneous protein crowding in liposomes: A new vista for the origin of cellular metabolism.

or

Souza et al. 2014, Journal of Molecular Evolution, Spontaneous encapsulation and concentration of biological macromolecules in liposomes: an intriguing phenomenon and its relevance in origins of life.

and in the same field a more recent study can be considered

Fanti et al., 2018 Integrative Biology: Do protocells preferentially retain macromolecular solutes upon division/fragmentation? A study based on the extrusion of POPC giant vesicles.

see also papers cited there for other examples. All these articles are unfortunately missed among the references of the present manuscript. The authors should at least mention them, although it would be very important to discuss the differences from and the novelty of the present work. Moreover, the phenomenon of the increase of membrane ion permeation by shear stress or by hydrostatic pressure is well known as reported in ref 36-38. Therefore the only novelty here, appear to be the application to this effect of the MANSION technology, already presented elsewhere refs 27.

Answer #1

We would like to thank the reviewer for his/her comments and the suggestion to clarify the novelty of our findings. In fact, the previous studies raised by the reviewer reported unpredicted concentrated situation of water-soluble macromolecules inside of liposomes when they were formed from lipid films. In our study, the preformed liposomes accumulated and condensed small molecules and ions upon the external flow of the aqueous solutions. Moreover, the liposomes repeated the hydrodynamic condensation (i.e. condensation by the solution of small molecules and ions, and following dilution by the solution which did not include them) without destruction and reformation of liposomal membrane. This non-equilibrium state of constrained liposomes upon the external flow was newly found in our MANSIONs. Therefore, we have revised the manuscript, especially abstract (p.1; rows 17-21) and introduction section (p.1; row 29-p.2, row 1, and p.2; rows 17-26) for clarifying the novelty of our findings mentioning the importance of the previous studies on super concentration effect which the reviewer has raised [ref.4,7,8]. In addition,

we modified the schematic illustration of MANSIONS in Fig. 1b to include the exchange process of the outer solution of the preformed and trapped liposomes.

4. de Souza, T. P., Fahr, A., Luisi, P. L. & Stano, P. Spontaneous Encapsulation and Concentration of Biological Macromolecules in Liposomes: An Intriguing Phenomenon and Its Relevance in Origins of Life. *J. Mol. Evol.* 79, 179-192, doi:10.1007/s00239-014-9655-7 (2014).
7. Luisi, P. L. et al. Spontaneous Protein Crowding in Liposomes: A New Vista for the Origin of Cellular Metabolism. *Chembiochem* 11, 1989-1992, doi:10.1002/cbic.201000381 (2010).
8. Fanti, A., Gammuto, L., Mavelli, F., Stano, P. & Marangoni, R. Do protocells preferentially retain macromolecular solutes upon division/fragmentation? A study based on the extrusion of POPC giant vesicles. *Integrative Biology* 10, 6-17, doi:10.1039/c7ib00138j (2018).

Comment #2- The phenomenon described here is mainly due to the far from equilibrium conditions which liposome are formed in or they are exposed to. In these cases, the real fate the lipid compartment follows, determines also the final state obtained. Therefore, compare the solute accumulation in liposome formed by the extrusion with that obtained by exposing lipid vesicles to a solute flow, it is for me conceptually wrong. In the first case, in fact, liposomes undergo a strong mechanical stress, in the second case, instead, just a hydro-dynamical perturbation. In my opinion, it is not really based on solid arguments to use this comparison, along with the absence of any observed accumulation by pipetting or vortexing liposomes in a solute solution, in order to state that this effect is driven by the interaction with a solid surface. (Rows 34-35) In fact, when dimensionally poly-dispersed giant vesicles, prepared by the swelling method, are forced to pass 10 times through a filter of 5um pores in presence of an external solute, the lipid membrane is broken and resealed many times. This process is completely different from exposing a preformed 20um extruded liposome to a solute flow. To assume that the mechanism of solute accumulation is comparable and based

on the surface interactions needs more solid argumentations.

Moreover, in the fig 2 extended Data liposomes are reported that have been vortexed and pipetted. Some of them in the enlarged image exhibit an inside region darker than the background or with a similar intensity that should suggest a solute accumulation or a similar concentration. Could you report on which compartments of the enlarged image the line profile have been determined by you or, alternatively, could you determine the line profile on those I have circled?

Moreover, why do you extruded on a different sizes pore filter the liposomes for the hydrodynamic pressure experiments, 20um instead of 5um? Please explain somewhere, since this make further incomparable these two procedures.

Answer #2

We would like to thank the reviewer for his/her critical comments. We agree with the reviewer's statement on the membrane destruction and recombination/reorientation during the filtration of narrow pores. To avoid confusing the readers, we have excluded the results on the filtration experiments of liposomes via 5 μm -pore-size filter in the Results section of the revised version (p.3; rows 14-30). We also agree with the reviewer's suggestion on Extended data Fig. 2 (shown in the previous version) that more careful discussion is needed for the comparison of the pipetting and vortexing experiments to the extrusion experiment. As stated above, in the revised manuscript, we excluded the results of the extrusion experiment. Even in this situation, the results of pipetting and vortexing experiments (these experiments did not provide the concentrated uranine inside of preformed cell-sized liposomes) can be recognized as an indispensable reference experiments to hypothesize that the condensation effect shown here was unique to the hydrodynamic conditions in the current microfluidic device. Thus, we replaced the results of the reference experiments of pipetting and vortexing using SDCM (Supplementary Fig. 9 in the revised version). In addition, the reviewer suggested to add the explicit identification of liposomes taken the line profile. Actually, the liposome taken the line profile was not shown in the low magnification images. To make the line profile precise, we used high-magnification lens to take the image of liposomes for taking their line profiles (the reason why we used the digitally enlarged image for the line profiles in the extrusion experiment was that taking the composite image of such small liposomes was technically difficult because of their Brownian motion). In addition, since uranine was easily photobleached in the SDCM observation, it was not preferable to observe same liposome accumulating uranine more than twice under the SDCM with different magnification lens. Therefore, we have revised the caption of the figure to clarify this point.

The reason why we extruded on 20 μm -pore-size filter the liposomes for their introduction into the trapping device was to avoid clogging in the microchannel. We have stated this in Methods section of the revised version (p.10; rows 40-42).

Comment #3- What is the exact composition of your liposomes? In the preparation procedure you stated that you swell a lipid film by approximately this composition 1:1:1:10 POPC/POPG/cholesterol/Fructose. Instead, in the main text (row 230) you declare that POPG is the main component of the membrane, while in the supplementary material (note for the quality of phospholipids) you show the NMR only of the POPC like the main component of the membrane. That's a bit confusing!! It is difficult to rationalize an effect without knowing which kind of compartments we are speaking on!!

Assuming that the membrane composition is 1:1:1 POPC/POPG/cholesterol, The different behaviour of the membrane transport the authors observed when liposomes are exposed to a solute flow can be rationalize as due to the negatively charged membrane that inhibits the passive diffusion of the negatively charged fluorescent molecules after the internal accumulation due to hydrostatic pressure.

It is not clear to me the role of fructose in the liposome preparation, the author should comment why this compound is necessary and why it is dried along with lipids in the film and not added to the water solution for the swelling, as normally it is done.

In my opinion, the authors should completely re-think this work focusing it only on the study of the accumulation of solute by hydrodynamic pressure by using the MANSION apparatus.

Answer #3

We agree with the reviewer's comment on the need to show the exact composition of liposome dispersion explicitly in the main manuscript. We have added the description for the preparation method and composition of the liposome dispersion as the first paragraph of the Results section (p.3; rows 6-13). In fact, as stated in our previous Method section, the concentration of POPC-stock solution (40 mM) was different from those of POPG- and cholesterol-stock solution (4.44 mM). Therefore, the composition of POPC/POPG/cholesterol/Fructose was 9/1/1/9 in the liposome dispersion in molar ratio. We also revised the related Method section to make it clear that the concentration stated is the concentration of the stock solution. The doping of fructose to lipid film had been reported elsewhere [ref.33] for obtaining spherical liposomes with low lamellarity.

As stated in Answer #2, we have focused on results for describing and clarifying the hydrodynamic condensation phenomena in the Results section of the revised version (p.3; rows 14-30).

Comment #4- I suggest to the authors to use the word 'accumulation' instead of 'condensation', since condensation is used in physical chemistry for describe the matter transformation from the liquid to the solid state with the formation of stronger bonds among molecules, instead accumulation is used in biological context to describe amount increase of different substrates in cells.

Answer #4

Thanks to the reviewer's comment, we have reconsidered the term for the current finding of unpredicted accumulation of the molecules in the liposomes explored by MANSIONS. We think that "accumulation" and "concentration" are appropriate representations to describe the process of concentration increase of encapsulated molecules which is to be applied in both cases along and against the Fick's law (concentration gradient-derived diffusion). So, we have revised "enrichment" as "accumulation" or "concentration" in the revised manuscript. However, we remark that the most striking result in the current manuscript is the achievement of the higher concentration of uranine and FL-ATP against the concentration gradient. We consider that these experimental results should be clearly distinguished to the results which can be derived from well-known diffusion across the membrane. "Accumulation" is not suitable to remark this point, and neither is "concentration", because it is also used as "the amount of a substance that is mixed with water or another substance". Thus, we do not change the use of "condensation" in the manuscript including the title.

Comment #5- Please somewhere at the beginning declare the composition of the membrane of your liposomes that could make the difference in the behaviour they exhibit.

Answer #5

We agree with the reviewer's comment. We have revised the first paragraph of the Results section (p.3; rows 6-13) for describing the preparation of liposomes and liposomal composition. The composition of POPC/POPG/cholesterol/Fructose was 9/1/1/9 in molar ratio.

Comment #6- Row 33- the meaning of sentence is not quite clear to me. What do the authors mean with "to assemble substances"? Maybe they mean: "to concentrate" or "to gather" substances.

Answer #6

We agree with the reviewer's comment. We have revised "to assemble substances" as "to concentrate substances" (p.2; row 14).

Comment #7- Rows 37-39, please insert a reference to justify the statement.

Answer #7

As to the revision of the Introduction section, we have referred additional papers [ref. 16, 17] for reinforcing the statements.

16. Melkikh, A. V. & Sutormina, M. Protocells and LUCA: Transport of substances from first physicochemical principles. *Prog. Biophys. Mol. Biol.* **145**, 85-104, doi:10.1016/j.pbiomolbio.2018.12.011 (2019).

17. Schmitt, C., Lippert, A. H., Bonakdar, N., Sandoghdar, V. & Voll, L. M. Compartmentalization and Transport in Synthetic vesicles. *Frontiers in Bioengineering and Biotechnology* **4**, 19, doi:10.3389/fbioe.2016.00019 (2016).

Comment #8- Row 45. Add references for MANSION.

Answer #8

Since we conceived and developed MANSIONs for this study as our original platform, no reference is necessary.

Comment #9- Row 56: Please explain while the membrane is fluorescent.

Answer #9

The membrane was fluorescent because the liposomes were prepared from lipid film containing Texas Red DHPE. We have added this information in the revised version (p.3, rows6-13).

Comment #10- Row 80: I have not well understood how you collected the liposomes for the images in Fig 2e,f and extended data fig. 3. In fact, the hydrodynamic pressure experiments are done with the MANSIONs equipment shown in fig. 1 where EFM is used.

Answer #10

The results shown in Fig. 2e, f and extended data Fig. 3 (in the previous version) were taken by not MANSIONs but manual handling of same trapping device. Since SDCM requires precise control of focus plain, SDCM observation cannot be performed automatically. In fact, SDCM is equipped to the same microscope which we used for MANSIONs (as the reviewer pointed out, this equipment description was not clear in the previous version). Thus, we have revised the main text (p3; row 23) and Method section (p.12; rows 40-43).

Comment #11- Row 87 Fig. 2 caption, Please add the pore size of the filter for both the cases in order to understand the different size of liposomes.

Answer #11

According to the revision of Fig.2 and the Results section, we have also revised the Fig.2 caption. In the revised version, it is not necessary to add the information of filtration the result of which is removed.

Comment #12- Row 109: Maybe 'slower' is better than 'smaller' speaking about the time of proton release.

Answer #12

We agree with the reviewer's comment. We have revised "smaller" as "slower" (p.4; row 23).

Comment #13- Row 234: The composition of the lipid film you form and swell by Millipore water before the extrusion has a composition 1:1:1:10 POPC/POPG/cholesterol/fructose are you sure that the POPG is the main component of your liposomes.

Answer #13

Please see Answer #3 and #5.

Comment #14- Extended Data. Fig. 2. Please add the size of the bar.

Answer #14

We have added the scale of the bar (Supplementary Fig. 9 caption). The scale bars represented 10 μm .

Reviewer #2

Comment #1- Q1: The condensation has been well-known already as shown in the following paper: J Mol Evol. 2014 Dec; 79(5-6):179-92. doi: 10.1007/s00239-014-9655-7. Epub 2014 Nov 22.

Spontaneous encapsulation and concentration of biological macromolecules in liposomes: an intriguing phenomenon and its relevance in origins of life. So what are the differences and innovations of this paper?

Answer #1

We would like to thank the reviewer for his/her comments and the suggestion to clarify the novelty of our findings. In fact, the previous studies raised by the reviewer reported unpredicted concentrated situation of water-soluble macromolecules inside of liposomes when they were formed from lipid films. In our study, the preformed liposomes accumulated and condensed small molecules and ions upon the external flow of the aqueous solutions. Moreover, the liposomes repeated the hydrodynamic condensation (i.e. condensation by the solution of small molecules and ions, and following dilution by the solution which did not include them) without destruction and reformation of liposomal membrane. This non-equilibrium state of constrained liposomes upon the external flow

was newly found in our MANSIONS. Therefore, we have revised the manuscript, especially abstract (p.1; rows 17-21) and introduction section (p.1; row 29-p.2, row 1, and p.2; rows 17-26)-sections for clarifying the novelty of our findings mentioning the importance of the previous studies on super concentration effect which the reviewer has raised [ref.4,7,8]. In addition, we modified the schematic illustration of MANSIONS in Fig. 1 to include the exchange process of the outer solution of the preformed and trapped liposomes.

4. de Souza, T. P., Fahr, A., Luisi, P. L. & Stano, P. Spontaneous Encapsulation and Concentration of Biological Macromolecules in Liposomes: An Intriguing Phenomenon and Its Relevance in Origins of Life. *J. Mol. Evol.* 79, 179-192, doi:10.1007/s00239-014-9655-7 (2014).
7. Luisi, P. L. et al. Spontaneous Protein Crowding in Liposomes: A New Vista for the Origin of Cellular Metabolism. *Chembiochem* 11, 1989-1992, doi:10.1002/cbic.201000381 (2010).
8. Fanti, A., Gammuto, L., Mavelli, F., Stano, P. & Marangoni, R. Do protocells preferentially retain macromolecular solutes upon division/fragmentation? A study based on the extrusion of POPC giant vesicles. *Integrative Biology* 10, 6-17, doi:10.1039/c7ib00138j (2018).

Comment #2- Q2: The infrequent condensation under the filter extrusion and the frequent condensation discovered in the micro-chamber may not be driven by the same mechanism. Please show the flow rate dependency of the infrequent condensation.

Answer #2

We would like to thank the reviewer for his/her critical comments. We agree with the reviewer's statement on the incomplete description between the infrequent condensation under the filter extrusion and the frequent condensation discovered in MANSIONS. It is certain that the membrane destruction and recombination/reorientation can occur during the filtration using a narrow size pore filter. Moreover, we approximately calculated the flow rate used in the extrusion experiment and found that it was ca 3000 times larger than that in the microfluidic experiment. Namely, the discussion on the direct comparison of these experiments in the previous version probably was incomplete. Therefore, to avoid confusing the readers, we have excluded the results on the filtration experiments of liposomes via 5 μm -pore-size filter in the Results section of the revised version (p.3; rows 13-30).

Comment #3- Q3: What is the main reason for non-equivalent kinetics of the intake and release of small molecules?

Answer #3

We postulate the reason for non-equivalent kinetics of the intake and release of small molecules as follows. When the trapped liposome is attached on the exit of the nest, the liposome is slightly forced to be deformed from spherical shape. Not only chemical

interaction of the PDMS surface of the exit but also the micrometer-scale stretching and buckling can cause reorientation and remodeling of phospholipid bilayer [B. Sani, et al, *NANO Lett.* 2008, 8, 866-871]. In case of heterogeneous phospholipid bilayer, a specific curvature resulted from strong bending derives the remodeling of lipid distribution even between outer and inner leaflets [P. V. Bashkirov, et al., *Biochemistry (Moscow) Supplementary Series A: Membrane and Cell Biology* 2011, 5, 205-211]. In this study, the liposome was also composed of a coupled of phospholipids. The trapped liposomes under the external flow plausibly occurred transient membrane disturbance, resulting in the unequal distribution of two phospholipids of different head groups between the outer and inner leaflets. Since recent computational researches revealed that membrane permeation of solutes and ions in aqueous phase is dependent of the interaction between lipid head group and solutes/ions [N. Pokhrel & L. Maibaum, *Journal of Chemical Theory and Computation*, 2018, 14, 1762-1771, M. A. Wilson & A. Pohorille, *Journal of the American Chemical Society*, 1996, 118, 6580-6587], the unequal distribution of phospholipids in the trapped liposomes primarily governs the non-equivalent kinetics of intake and release of small molecules/ions. Along to this context, we have tried to trap liposomes of POPC, which were prepared in the same method, and measure them by MANSIONS. But, the surface of the fluidic device was remarkably adhesive to the liposomes of POPC, and their size was smaller than that of the liposomes described in main text. Namely, the postulation here was not evaluated in the current design of the trapping device. Thus, the time and effort necessary to find direct experimental evidences are at least beyond us in terms of our current setup of MANSIONS. We are considering working on this issue in our future research.

Comment #4- Q4: This paper should be improved by including direct experimental evidence to support the mechanism for the condensation speculated as follow:

“The outer leaflet of the trapped liposome is disturbed by contacting with a substrate, while the inner leaflet is kept rather intact. Thus, the energetic barrier for the partitioning could be asymmetric between the inside and outside, resulting in hydrodynamic condensation.”

Answer #4

We agree that it would be useful and fascinating that direct experimental evidences support the mechanism for the condensation speculated. As discussed in Answer #3, the clue of the non-equivalent kinetics of the intake and release of small molecules and ions is maybe associated with the remodeling of phospholipid molecules within each liposome. Then we observed laurdan-containing liposomes (0.2 mol% to lipids) introduced and trapped in the trapping device by EFM (excitation: 330–385 nm; emission: >420 nm (long path filter)). We compared the images taken under flow condition of 40 $\mu\text{L}/\text{h}$ and ca. 0 $\mu\text{L}/\text{h}$ respectively. As a result, in both cases no mosaic pattern of the fluorescence image of the trapped liposomes was observed even upon the external flow. This result indicates that the hydrodynamic condensation was not caused by the lateral phase separation in micrometer scale. As to asymmetric remodeling of phospholipids between inner and outer

leaflets, we focused on the liposomes of asymmetric membrane which is generated by water-in-oil emulsion transfer method [Pautot, S.; Frisken, B. J.; Weitz, D. A., *Proc. Natl. Acad. Sci. U. S. A.* 2003, 100 (19), 10718-10721.]. However, the liposomes prepared by the water-in-oil emulsion transfer method strongly adhered to the surface of the microfluidic device and was not applicable to the current setup of MANSIONS. Thus, the time and effort necessary to find direct experimental evidences are at least beyond us in terms of our current setup of MANSIONS. We are considering working on this issue in our future research.

Thanks to the reviewer's comment, during the revision process, we have found that our previous statement "Thus, the energetic barrier for the partitioning could be asymmetric between the inside and outside, resulting in hydrodynamic condensation." contained specific speculation and was not suitable. We have thus revised the sentence as "Such degree of the membrane disturbance might cause the change of partitioning. Thus, the partitioning kinetics could be asymmetric between the inside and outside, resulting in hydrodynamic condensation." (p.9; rows 32-34)

Reviewer #3

Comment #1- The only weak point in this paper, in my opinion, is the statistical discussion of the data, which is very limited. Since the Authors claim that the condensation is an infrequent event, one would expect also a discussion about the statistical distribution of the filled liposomes vs empty liposomes. Is this a power-law distribution as it happens for another spontaneous liposome filling, like the "super crowding effect"? (about this, see: Luisi, Pier Luigi, et al. "Spontaneous protein crowding in liposomes: A new vista for the origin of cellular metabolism." *ChemBioChem* 11.14 (2010): 1989-1992.). I suggest the Authors to integrate the paper with a statistical discussion and with a comparison between their results about the condensation of small molecules and the supercrowding of high-molecular weight proteins described in the above cited paper.

Answer #1

We would like to thank the reviewer for his/her comments and the suggestion to clarify the novelty of our findings, especially in the view of statistical distribution of liposomes. First of all, we have revised the manuscript to more clearly explain the general novelty of our results. In fact, the previous studies raised by the reviewer reported unpredicted concentrated situation of water-soluble macromolecules inside of liposomes when they were formed from lipid films. In our study, the preformed liposomes accumulated and condensed small molecules and ions upon the external flow of the aqueous solutions. Moreover, the liposomes repeated the hydrodynamic condensation (i.e. condensation by the solution of small molecules and ions, and following dilution by the solution which did not include them) without destruction and reformation of liposomal membrane. This non-equilibrium state of constrained liposomes upon the external flow was newly found in our MANSIONS. Therefore we have revised the manuscript, especially abstract (p.1; rows 17-21) and introduction section (p.1; row 29-p.2, row 1, and p.2; rows 17-26)-sections for clarifying the novelty of our findings mentioning the importance of the previous studies on

super concentration effect which the reviewer has raised [ref.4,7,8]. We also modified the schematic illustration of MANSIONS in Fig. 1b to include the exchange process of the outer solution of the preformed and trapped liposomes.

Regarding to the reviewer's suggestion on the statistical viewpoint, we replotted the histogram of liposomes condensing uranine inside shown in Supplementary Fig. 13 to the double logarithmic plot. A negative correlation between the GFI_{lipo} and its frequency was observed for the liposomes of high concentration of uranine, while the correlation was not clear for the liposomes of low concentration of uranine and their frequency. However, it is difficult to remark the novelty or similarity of our results in statistical aspect, e.g. the shape of the distribution, regarding a power law as shown in the studies by Luisi's group.

On the other hand, we have revised the ambiguous statement for the statistics of liposomes such as "some/most liposomes" as "xx% of trapped liposomes" (p.4; row 13, p4; row 32-p.5; row 1, p.8; row 8).

4. de Souza, T. P., Fahr, A., Luisi, P. L. & Stano, P. Spontaneous Encapsulation and Concentration of Biological Macromolecules in Liposomes: An Intriguing Phenomenon and Its Relevance in Origins of Life. *J. Mol. Evol.* 79, 179-192, doi:10.1007/s00239-014-9655-7 (2014).
7. Luisi, P. L. et al. Spontaneous Protein Crowding in Liposomes: A New Vista for the Origin of Cellular Metabolism. *Chembiochem* 11, 1989-1992, doi:10.1002/cbic.201000381 (2010).
8. Fanti, A., Gammuto, L., Mavelli, F., Stano, P. & Marangoni, R. Do protocells preferentially retain macromolecular solutes upon division/fragmentation? A study based on the extrusion of POPC giant vesicles. *Integrative Biology* 10, 6-17, doi:10.1039/c7ib00138j (2018).

Reviewers' comments:

Reviewer #1 (Remarks to the Author):

Although the manuscript is now much clear and easily readable, in my opinion there are some points that must be checked before to publish it.

The first point is there is not a convincing evidence of a real hydrodynamic accumulation of uranine, please not 'condensation' that is a misleading term, inside GVs. Indeed, the authors would have had to work with a pH-buffered solution, to have a fine control of the external and the internal pH of giant vesicles. This is a crucial point if one works with a fluorescent probe that is pH dependent with an inflection point in the GFI/pH curve around 7.0: the pH value of a neutral solution.

The authors in the material and methods section do not indicate any pH buffer in the vesicles preparation.

The accumulation, they claim, is experimentally evidenced by an increase in the fluorescence intensity inside trapped liposomes that could be due to higher pH inside the lipid compartment. In this case, there might be no accumulation but only a different fluorescence of the probe at the same concentration, but in different pH conditions.

Since the external pH measured is around 6.2 it is enough an internal pH around 7.5 to observe an increase in the fluorescence intensity going for 50 to 200au. It does not seem a convincing proof of avoiding an overvaluation of the uranine concentration inside liposomes due to an unbalanced pH, to report the ratio $(GFI)_{lipo}/(GFI)_{BG} > 20$, if the $(GFI)_{BG}$ tends to zero, i.e. near to limit of the signal to noise ratio, when both pH and the uranine concentration (0.5uM) decrease.

On the other hand, the experiments done with BMeS-p-A (2,5-bis(methylsulfonyl)-1,4-phenylenediamine), a dye which fluorescence emission does not depend on pH, show no solute accumulation, i.e. no difference in the fluorescence intensity inside and outside the liposome. Moreover, when the sample of GVs with higher uranine intensity is washed with a 10 μ M HCl/5 μ M uranine/1 mM fructose solution (pH 4.68) the intensity of 70% GVs was quenched, and this could be explained by the pH re-equilibration inside and outside the vesicles. Thus, these vesicles do not exhibit longer any increase of the internal fluorescence for a further hydrodynamic perturbation, like authors observed. Only 30% of vesicles keep their higher internal fluorescence, that is only 3 liposomes.

The experiments with fluorescein-tagged ATP suffer of the same critical point, in my opinion. So my suggestion is to work with a pH buffered solution of uranine around a physiological value to avoid any doubts, or please explain why you do not do this kind of simple experiments, at least as a control experiment.

Concerning the theoretical calculations, the model for the diffusion of charged solutes, like protons and uranine, cannot be based on the simple Fick's law that is suitable for uncharged molecules. For charged compounds, the electrostatic potential that takes place across the membrane must be taken into account along with the migration of counter ions. Please discuss better your approximations.

Finally, I would like to remark, the liposomes the authors used in this work:

POPC/POPG/Cholesterol 9/1/1 are not very plausible like primordial compartments (see. Ruiz-Mirazo Chem. Rev. 2014, 114, 285–366) instead, "alkyl phosphates, alkyl sulfates, fatty acids, and polyprenyl chains have been proposed as possible constituents of early membranes". Indeed, these single chain amphiphiles are less robust of double chain phospholipids and their behaviour under hydrodynamic perturbation must be checked. Therefore, in drawing the conclusions, the authors should keep in mind this point.

Also the role of the fructose, present at 1mM concentration, too high for a primordial scenario, it is not completely clear and should be better discussed. What does it happen, if fructose is not present? Of course, it is worthwhile to clarify the point, if authors want to stress the hydrodynamic perturbation as a possible mechanism of solute accumulation in prebiotic scenario.

In conclusion, in my opinion, the hydrodynamic perturbation can speed up the passive transport of molecule across the lipid membrane, and this could have had some role in a primordial scenario. But this intriguing phenomenon need a more extensive study to be elucidated, since it appear to

be dependent on the molecular structures and, maybe, on the electric charge of the solute. I cannot see a convincing proof of molecular accumulation in this manuscript, without any doubts, and it does not appear a so general behaviour as claimed by the authors.

Reviewer #2 (Remarks to the Author):

The authors did not answer my questions 3 and 4 directly. This means that no mechanism for what authors found, the hydrodynamic condensation of small molecules and ions into cell-sized liposomes, is available. Putting it the other way around, their finding is an "interesting" phenomenon which occurs only in their experimental setup, MANSION. Without a plausible mechanism supported by experimental evidence, other researchers would hardly extend what the authors found. Therefore, I do not think that the current manuscript contains contents sufficient for the publication to Communications Chemistry.

Reviewer #3 (Remarks to the Author):

The revised version takes into account all the remarks I have rised in the original version, therefore in my opinion the present paper is acceptable for publication.

Point-by-point response sheet

Reviewer #1

Comment #1

Please not 'condensation' that is a misleading term, inside GVs.

Answer #1

Reviewer #1 kindly suggested the use of “accumulation” instead of “condensation” on our findings. We agree to the reviewer’s suggestion. To remark that the concentration of small molecules and ions inside of liposomes was higher than that of the outside, in the revised version including the title, we have used “accumulation” of the substances with the phrase of “against concentration gradient” across the liposomal membrane as far as the readability is not dismissed.

Comment #2

The authors would have had to work with a pH-buffered solution, to have a fine control of the external and the internal pH of giant vesicles. This is a crucial point if one works with a fluorescent probe that is pH dependent with an inflection point in the GFI/pH curve around 7.0: the pH value of a neutral solution.

... So my suggestion is to work with a pH buffered solution of uranine around a physiological value to avoid any doubts, or please explain why you do not do this kind of simple experiments, at least as a control experiment.

Answer #2

We appreciate the reviewer’s suggestion on the need for the use of a buffer solution. We have added the data on the Supplementary figure 11. In fact, the liposomes GFI_{lipo} of which were larger than GFI_{BG} were also observed even when the liposomes were formed by a buffer solution (pH 7.87), trapped, and exposed to a buffer solution (pH 7.87) containing 5 μ M uranine. We thus revised the main text describing the data on the hydrodynamic accumulation of uranine. The reason why we initially avoid using buffer solution was to keep the variety of experimental materials small. As reviewer #1 pointed out, our findings certainly involve complicated pathways of the transportation of charged molecules across the liposomal membrane. That is why we have piled the quantitative and circumstantial proofs

for the encapsulation and high concentration of uranine inside (we also discuss this in Answers #3, #4, #6, and #7 and made a major revision of the main text).

Comment #3

Since the external pH measured is around 6.2, it is enough an internal pH around 7.5 to observe an increase in the fluorescence intensity going for 50 to 200 au. It does not seem a convincing proof of avoiding an overvaluation of the uranine concentration inside liposomes due to an unbalanced pH, to report the ratio $(GFI)_{\text{lipo}}/(GFI)_{\text{BG}} > 20$, if the $(GFI)_{\text{BG}}$ tends to zero, i.e. near to limit of the signal to noise ratio, when both pH and the uranine concentration (0.5uM) decrease.

Answer #3

We accept the concern of reviewer #1 on the overvaluation of inner concentration of uranine caused by the pH gradient. Since the details on this issue were partly omitted in the previous version, reviewer #1 may be disappointed. Thus, we have revised the main text to clearly answer to this issue in the revised version.

In fact, the distribution of GFI_{BG} of nests where the 0.5 μM uranine solution was added was moderately fitted by the Gaussian distribution (average 0.71, standard deviation; 0.17; Supplementary Figure 11). Therefore, the probability to obtain $GFI_{\text{lipo}}/GFI_{\text{BG}} > 17.8$ (smallest value among $GFI_{\text{lipo}}/GFI_{\text{BG}}$ of the liposomes fluorescing brilliantly at 3300 seconds in Fig. 3d) was smaller than 0.05% even assuming the probably highest concentration gradient for one measurement. We notice that the probability to obtain $GFI_{\text{lipo}}/GFI_{\text{BG}} > 17.8$ was much smaller than 0.05% in practice, because we repeated three experiments with the same protocol and the number of data to be justified was over 30. Moreover, the distribution of $GFI_{\text{lipo}}/GFI_{\text{BG}}$ was in the range from 17.8 to over 100, and its averaged value was 37.8. It is not realistic to justify these large values of $GFI_{\text{lipo}}/GFI_{\text{BG}}$ by only the pH gradient assuming equal concentration of uranine inside and outside.

Therefore, even though there were slight overvaluations of the uranine concentration in terms of $GFI_{\text{lipo}}/GFI_{\text{BG}}$, we can deduce that the current data is enough to point out that the concentration of uranine inside of the trapped liposomes was higher than that of outside upon the uranine exposure.

Comment #4

On the other hand, the experiments done with BMeS-p-A (2,5-bis(methylsulfonyl)-1,4-phenylenediamine), a dye which fluorescence emission does

not depend on pH, show no solute accumulation, i.e. no difference in the fluorescence intensity inside and outside the liposome.

Answer #4

The reason why BMeS-p-A was not accumulated in the liposomes is not clear currently. However, as we have explained in Answer #3, the difference between uranine and BMeS-p-A in terms of the hydrodynamic accumulation is not misidentified by the effect of pH gradient. The selectivity of small molecules able to accumulate in trapped liposomes will be explored regard to their chemical structures and properties in future.

Comment #5

Moreover, when the sample of GVs with higher uranine intensity is washed with a 10 μ M HCl/5 μ M uranine/1 mM fructose solution (pH 4.68) the intensity of 70% GVs was quenched, and this could be explained by the pH re-equilibration inside and outside the vesicles. Thus, these vesicles do not exhibit longer any increase of the internal fluorescence for a further hydrodynamic perturbation, like authors observed. Only 30% of vesicles keep their higher internal fluorescence, that is only 3 liposomes.

Answer #5

We thank the reviewer for the careful discussion on the experiment using acidic uranine solution. First of all, the purpose of this experiment was not demonstration of hydrodynamic accumulation of uranine the concentration of which was higher than that of outside of liposomes, but reliable confirmation of encapsulation of uranine in the liposomes.

The line profiles analyzed from the SDCM images of all the liposomes had same tendency in terms of uranine (unimodal and convex) and liposomal membrane (bimodal and concave) (Supplementary Fig. 7). These results have already indicated the existence of uranine inside of liposomes. We would reinforce the experimental result in order to strongly indicate the encapsulation of uranine. Namely, the introduction of acidic uranine solution to liposomes leads spatial distinction of the outside and inside of the membrane. Our aim was thus to obtain the time delay of the decrease of GFI_{lipo} comparing with GFI_{BG} . The delay of quench of uranine inside of liposomes does not occur without the shell of liposomal membrane (Fig. 3a, 3b). On the contrary, for liposomes the fluorescence intensity of which diminished instantaneously (Fig. 3c), uranine could had been adhered to liposomal membrane somehow during the uranine exposure, or uranine had been certainly encapsulated inside but uranine eluded or protons instantaneously intruded when the acidic uranine solution was

introduced. Considering the results of SDCM observation, it is plausible that, even in case of the liposomes described in Fig. 3c, the uranine had been encapsulated in the trapped liposomes and protons intruded during the addition of the acidic uranine solution. Therefore, we do not think that data and discussion in the main text was in need of crucial reconsideration.

Comment #6

The experiments with fluorescein-tagged ATP suffer of the same critical point, in my opinion.

Answer #6

Justifying statements on the hydrodynamic accumulation of FL-ATP inside of liposomes were essentially different from that of uranine. The important point was the non-monotonic time course of GFI_{lip0} caused by the acidity of FL-ATP itself:

First, the recovery of fluorescence intensity of FL-ATP during the washout (30–60 min) suggested the continuous encapsulation of FL-ATP during 5–20 min. Second, the strength of suppression of fluorescence from FL-ATP was positively related to the amount of FL-ATP. Third, at a couple of liposomes, GFI_{lip0} was higher than GFI_{BG} at 5 min although at 10 min the fluorescence intensity of FL-ATP inside the liposome was more strongly suppressed than the outer solution. Therefore, the concentration of FL-ATP inside liposomes upon the FL-ATP exposure was higher than that of the outer FL-ATP solution.

We have revised the main text with these justifying statements.

Comment #7

Concerning the theoretical calculations, the model for the diffusion of charged solutes, like protons and uranine, cannot be based on the simple Fick's law that is suitable for uncharged molecules. For charged compounds, the electrostatic potential that takes place across the membrane must be taken into account along with the migration of counter ions. Please discuss better your approximations.

Answer #7

We thank the reviewer for the thoughtful indication on the theoretical handling of the charged solutes. We reconsidered the model by explicitly adopting the Donnan potential caused by the immobilized charged phospholipid (especially POPG). As a result, we found that one of heuristic assumptions required in the former simulation can be omitted to

reproduce the time course of GF_{lip0} , and the representation of the simulation got much clearer.

On the other hand, for the estimation of the permeability of the proton was originally proceeded as not only a proof of the existence of uranine inside of liposomes (Answer #5) but also an example to show the low permeability of smaller molecules and ions to investigate the fundamental process of the permeation in this experimental system. Since our discussion was largely revised with the additional data (Supplementary Fig. 19–21), we now believe that the discussion on the permeation of the proton was not necessary in our manuscript.

Comment #8

Finally, I would like to remark, the liposomes the authors used in this work: POPC/POPG/Cholesterol 9/1/1 are not very plausible like primordial compartments (see. Ruiz-Mirazo, Chem. Rev. 2014, 114, 285–366) instead, “alkyl phosphates, alkyl sulfates, fatty acids, and polyprenyl chains have been proposed as possible constituents of early membranes”. Indeed, these single chain amphiphiles are less robust of double chain phospholipids and their behaviour under hydrodynamic perturbation must be checked. Therefore, in drawing the conclusions, the authors should keep in mind this point.

Answer #8

We appreciate reviewer’s insight and suggestion about the plausible membrane molecule candidates at the earliest stage of protocells. We revised our manuscript considering this point.

It is interesting whether compartments composed of simpler amphiphiles afford the hydrodynamic accumulation of small molecules and ions. However, such experiments are beyond our current purpose, and the fundamental viewpoint raised in our manuscript was how to make the benefit and backwash of the compartmentalization compatible, which would be critical especially for the phospholipid membrane of lower permeability. Since at some stage of the evolution toward the current living cells, the cells started to utilize phospholipid membrane, our findings bring new insight.

Comment #9

Also the role of the fructose, present at 1mM concentration, too high for a primordial scenario, it is not completely clear and should be better discussed. What does it happen, if

fructose is not present? Of course, it is worthwhile to clarify the point, if authors want to stress the hydrodynamic perturbation as a possible mechanism of solute accumulation in prebiotic scenario.

Answer #9

We agreed with the reviewer's comment and added the data obtained by liposomes prepared without fructose (Supplementary Fig. 19). As we stated in the previous response sheet and revised the former version, the addition of fructose was for preparing liposomes with lower lamellarity [ref. 34]. In fact, multilamellar liposomes tended to clog the bypass path in trapping region because of their low deformability or structural stability against the rupture, which was problematic for the trapping mechanism of the nest structure. Thus, the use of fructose is rather the technical requirement. Since the data in case of liposomes without fructose was comparable to that obtained by liposomes prepared with fructose, we do not think that the use of fructose is not problematic to claim that the hydrodynamic accumulation of small molecules and ions across the liposomal membrane could play a potential role on the origin of life.

Comment #10

In conclusion, in my opinion, the hydrodynamic perturbation can speed up the passive transport of molecule across the lipid membrane, and this could have had some role in a primordial scenario. But this intriguing phenomenon need a more extensive study to be elucidated, since it appear to be dependent on the molecular structures and, maybe, on the electric charge of the solute. I cannot see a convincing proof of molecular accumulation in this manuscript, without any doubts, and it does not appear a so general behaviour as claimed by the authors.

Answer #10

We sincerely thank to the reviewer's careful reading and discussion to improve our manuscript. In former version of the manuscript, the mechanism explanation might be confusing as if it were applicable to the general situation. We would again raise the list of the major revision in the current version as follows.

- ✓ We added experimental results of the hydrodynamic accumulation of uranine prepared in absence of POPG or fructose. In case of liposomes prepared without POPG, the hydrodynamic accumulation was suppressed, while the removal of fructose had no effect. According to the results, we revised the discussion part as well as Methods.

- ✓ We added zeta potential data of liposome dispersions to discuss about the membrane surface charge. According to the results, we revised the discussion part as well as Methods.
- ✓ We examined the hydrodynamic accumulation with buffer solution and found that uranine was accumulated in liposomes. According to the results, we revised the discussion as well as Methods.
- ✓ As a result, in revision, we concentrated the explanation on the mechanism of hydrodynamic accumulation against the concentration gradient in terms of surface charge of liposomes and dependence on the solute species.

Now we believe that the discussion and conclusion in the current version is crucially improved according to the reviewer's comments.

Reviewer #2

Comment #1

The authors did not answer my questions 3 and 4 directly. This means that no mechanism for what authors found, the hydrodynamic condensation of small molecules and ions into cell-sized liposomes, is available. Putting it the other way around, their finding is an "interesting" phenomenon which occurs only in their experimental setup, MANSION. Without a plausible mechanism supported by experimental evidence, other researchers would hardly extend what the authors found.

- Question #3: What is the main reason for non-equivalent kinetics of the intake and release of small molecules?
- Question #4: This paper should be improved by including direct experimental evidence to support the mechanism for the condensation speculated as follow:
"The outer leaflet of the trapped liposome is disturbed by contacting with a substrate, while the inner leaflet is kept rather intact. Thus, the energetic barrier for the partitioning could be asymmetric between the inside and outside, resulting in hydrodynamic condensation."

Answer #1

We sincerely appreciate for the reviewer's comment with the integrity on the further extension on this phenomenon by other researchers. According to the comments, we added the hydrodynamic accumulation examination on the liposomes prepared without POPG and

measurement on the zeta potential of liposomes with and without POPG. As a result, we found that POPG could be the key to cause the hydrodynamic accumulation against the concentration gradient (only 1 of 15 of liposomes showed larger GFI_{lipo} for liposomes without POPG; Supplementary Fig. 20). In addition, it was experimentally evidenced that the addition of 10 mol% of POPG strikingly affect the zeta potential of liposomes (Supplementary Fig. 21).

Thus, currently we can postulate that the mechanism of the hydrodynamic accumulation of substances across the liposomal membrane against the concentration gradient is an asymmetric distribution of POPG between inner and outer leaflets. When the liposome was imposed to the surface of the device upon the external flow, the contact caused transient negative curvature and some extent of membrane perturbation, which might be a critical process to bias the distribution of POPG. As a result, the surface charge of the liposome might be imbalanced between inside and outside of the liposomal membrane. The permeation of the small molecules and ions through partitioning process was influenced by the surface charge of the membrane, and then, the entrapped substances were more strongly inhibited to permeate outward.

Although there are still unveiled mechanism, the key factors were now clarified, and current estimation becomes open to the further examinations and extension widely for researchers in soft matter physics, supramolecular chemistry, and synthetic biology.

REVIEWERS' COMMENTS:

Reviewer #1 (Remarks to the Author):

The revised manuscript takes into account most of the remarks I have raised. Nevertheless, I am convinced that the hydrodynamic solute accumulation requires more investigations to be elucidate in details. Indeed, this phenomenon appears to be strongly dependent both on the composition of the lipid membrane and on the molecular structure of the solute, to be considered as a quite general route for increasing the molecular amount inside lipid vesicles. Moreover, the explanation the authors offer to account for the observed data, i.e. the asymmetric composition of charged lipids in the vesicle membrane, does not convince me completely.

On the other hand, this paper has the merit to introduce this topic to the scientific community, and this could stimulate more researches in this direction. Therefore, in my opinion, the present paper is acceptable for publication, but I would like to suggest the authors to better stress, in the final discussion, that the present manuscript represents just a first step in the study of this puzzling process.

Point-by-point response sheet

Reviewer #1

Comment #1

The revised manuscript takes into account most of the remarks I have raised. Nevertheless, I am convinced that the hydrodynamic solute accumulation requires more investigations to be elucidate in details. Indeed, this phenomenon appears to be strongly dependent both on the composition of the lipid membrane and on the molecular structure of the solute, to be considered as a quite general route for increasing the molecular amount inside lipid vesicles. Moreover, the explanation the authors offer to account for the observed data, i.e. the asymmetric composition of charged lipids in the vesicle membrane, does not convince me completely.

On the other hand, this paper has the merit to introduce this topic to the scientific community, and this could stimulate more researches in this direction. Therefore, in my opinion, the present paper is acceptable for publication, but I would like to suggest the authors to better stress, in the final discussion, that the present manuscript represents just a first step in the study of this puzzling process.

Answer #1

We appreciate the reviewer thus far for advice improving our manuscript. We agree with the reviewer pointing out the importance of the statement avoiding the exaggerated claims from our manuscript. Thus, in the last paragraph in the main text, we revised the sentence "It can provide a clue to consider a molecular scenario for the continuous development of protocell in the early earth by suggesting a promising combination of molecular species and compartment enabling to earn and retain substances repeatedly through the phospholipid membrane." as "Although full elucidation of this puzzling permeation process including the mechanism would require further researches, the observed molecular dependencies can provide a clue to consider a molecular scenario for the continuous development of protocell in the early earth by suggesting a promising combination of molecular species and compartment enabling to earn and retain substances repeatedly through the phospholipid membrane."